EMBO
Molecular Medicine

# Striatal infusion of cholesterol promotes dose-dependent behavioral benefits and exerts disease-modifying effects in Huntington's disease mice

Giulia Birolini[1,2,‡], Marta Valenza[1,2,‡,*], Eleonora Di Paolo[1,2,‡], Elena Vezzoli[1,2,†], Francesca Talpo[3], Claudia Maniezzi[3], Claudio Caccia[4], Valerio Leoni[5], Franco Taroni[4], Vittoria D Bocchi[1,2], Paola Conforti[1,2], Elisa Sogne[6], Lara Petricca[7], Cristina Cariulo[7], Margherita Verani[7], Andrea Caricasole[7], Andrea Falqui[6], Gerardo Biella[3] & Elena Cattaneo[1,2,**]

## Abstract

A variety of pathophysiological mechanisms are implicated in Huntington's disease (HD). Among them, reduced cholesterol biosynthesis has been detected in the HD mouse brain from pre-symptomatic stages, leading to diminished cholesterol synthesis, particularly in the striatum. In addition, systemic injection of cholesterol-loaded brain-permeable nanoparticles ameliorates synaptic and cognitive function in a transgenic mouse model of HD. To identify an appropriate treatment regimen and gain mechanistic insights into the beneficial activity of exogenous cholesterol in the HD brain, we employed osmotic mini-pumps to infuse three escalating doses of cholesterol directly into the striatum of HD mice in a continuous and rate-controlled manner. All tested doses prevented cognitive decline, while amelioration of disease-related motor defects was dose-dependent. In parallel, we found morphological and functional recovery of synaptic transmission involving both excitatory and inhibitory synapses of striatal medium spiny neurons. The treatment also enhanced endogenous cholesterol biosynthesis and clearance of mutant Huntingtin aggregates. These results indicate that cholesterol infusion to the striatum can exert a dose-dependent, disease-modifying effect and may be therapeutically relevant in HD.

**Keywords** aggregates; behavior; cholesterol; Huntington's disease; synapses
**Subject Categories** Genetics, Gene Therapy & Genetic Disease; Neuroscience

## Introduction

The brain is the most cholesterol-rich organ, accounting for about 25% of the whole body's cholesterol (Dietschy & Turley, 2004; Dietschy, 2009). Its biosynthesis occurs through a stepwise cascade involving several enzymes under transcriptional regulation by sterol regulatory element binding protein 2 (SREBP2) transcription factor (Seo *et al*, 2012). In the adult brain, a small amount of cholesterol continues to be synthesized locally, where it regulates multiple processes including synapse formation and maintenance, synaptic vesicle (SV) recycling, and optimal release of neurotransmitters for downstream intracellular signaling pathways (Mauch *et al*, 2001; Rohrbough & Broadie, 2005; Fukui *et al*, 2015; Postila & Róg, 2020). Consequently, dysregulation of brain cholesterol homeostasis is linked to several chronic neurological and neurodegenerative diseases (Valenza & Cattaneo, 2011; Martin *et al*, 2014). Among these conditions is Huntington's disease (HD), an adult-onset disorder characterized by motor, cognitive, and psychiatric features.

The basis of HD is expansion of a CAG trinucleotide repeat in the gene encoding the Huntingtin protein (HTT; Saudou & Humbert, 2016). In HD, the striatal medium spiny neurons (MSNs) and

1   Department of Biosciences, University of Milan, Milan, Italy
2   Istituto Nazionale di Genetica Molecolare "Romeo ed Enrica Invernizzi", Milan, Italy
3   Department of Biology and Biotechnologies, University of Pavia, Pavia, Italy
4   Unit of Medical Genetics and Neurogenetics, Fondazione I.R.C.C.S. Istituto Neurologico Carlo Besta, Milan, Italy
5   School of Medicine and Surgery, Monza and Laboratory of Clinical Pathology, Hospital of Desio, ASST-Monza, University of Milano-Bicocca, Milan, Italy
6   Biological and Environmental Science & Engineering (BESE) Division, NABLA Lab, King Abdullah University of Science and Technology (KAUST), Thuwal, Saudi Arabia
7   Neuroscience Unit, Translational and Discovery Research Department, IRBM S.p.A, Rome, Italy
    *Corresponding author. Tel: +39 02 50325851; E-mail: marta.valenza@unimi.it
    **Corresponding author. Tel: +39 02 5032 5846; E-mail: elena.cattaneo@unimi.it
    †Present address: Department of Biomedical Sciences for Health, University of Milan, Milan, Italy
    ‡These authors contributed equally to this work

cortical pyramidal neurons projecting to the striatum are primarily affected and degenerate (Zuccato *et al*, 2010; Rüb *et al*, 2016). One of the underlying pathophysiological mechanisms is disruption of brain cholesterol biosynthesis due to reduced nuclear translocation of SREBP2 and diminished expression of its downstream target genes in the cholesterol biosynthesis pathway (Valenza *et al*, 2005, 2015a; Di Pardo *et al*, 2020). This defect occurs in astrocytes, which are the major producers of cholesterol in the adult brain, with consequent reduction of newly synthetized cholesterol available for neuronal function (Valenza *et al*, 2010, 2015a). This compromised process is manifested in reduced mRNA levels of key enzymes in the cholesterol synthesis pathway in HD cells, HD mouse models, and post-mortem brain from HD patients (Sipione *et al*, 2002; Valenza *et al*, 2005, 2015a; Bobrowska *et al*, 2012; Lee *et al*, 2015). Also evident is a reduction in cholesterol precursors, in particular lathosterol and lanosterol, as judged by isotopic dilution mass spectrometry (ID-MS) in HD cells (Ritch *et al*, 2012) and in seven rodent HD models (Valenza *et al*, 2007a,b, 2010; Shankaran *et al*, 2017); moreover, striatal level of lathosterol is inversely correlated with CAG repeat number (Shankaran *et al*, 2017). In addition, the striatum of knock-in zQ175 mice (Shankaran *et al*, 2017) shows reduced synthesis of new cholesterol after administration of deuterated water *in vivo* at the pre-symptomatic stage (Shankaran *et al*, 2017) and decreased cholesterol at late time points, as measured by gas chromatography–mass spectrometry (GC-MS). Finally, levels of the brain-derived cholesterol catabolite 24S-hydroxycholesterol (24S-OHC) are decreased in brain and blood from HD mice (Valenza *et al*, 2007b, 2010; Shankaran *et al*, 2017), post-mortem caudate (Boussicault *et al*, 2016), plasma of HD patients (Leoni *et al*, 2008, 2011), as well as in pre-HD manifesting patients who are close to the disease onset (Leoni *et al*, 2013).

All of these findings support the idea that the primary event of cholesterol dysfunction in the HD brain is decreased synthesis, followed by reduced conversion of cholesterol into 24S-OHC, ultimately leading to reduced cholesterol content at the late-symptomatic stages (Shankaran *et al*, 2017). In two studies conducted in two HD animal models (R6/2 and zQ175 mice), enhancement of cholesterol catabolism in the striatum via neuronal overexpression of cholesterol 24-hydroxylase (Cyp46A1) increased striatal levels of 24S-OHC and endogenous cholesterol biosynthesis and rescued several disease phenotypes (Boussicault *et al*, 2016; Kacher *et al*, 2019). Despite conflicting data regarding the amount of striatal cholesterol measured in the two studies in R6/2 and zQ175 adult mice (Boussicault *et al*, 2016; Kacher *et al*, 2019), this strategy proved to be of therapeutic relevance in targeting cholesterol biosynthesis in HD brain.

We previously reported that systemic administration of brain-permeable polymeric nanoparticles loaded with cholesterol (g7-NPs-chol) reversed synaptic alterations and prevented cognitive defects in a HD transgenic mouse model (Valenza *et al*, 2015b). This work provided the first proof-of-concept that cholesterol delivery to the HD brain is beneficial, but the low cholesterol content of g7-NPs-chol did not allow for full characterization of a target therapeutic dose or its effects on motor and cognitive capacity.

Here we sought to evaluate the dose-dependent effects of cholesterol infusion on molecular, functional, and behavioral parameters. For this purpose, we took advantage of osmotic mini-pumps to infuse three escalating doses of cholesterol directly into the striatum of HD mice, in a continuous and rate-controlled manner. In this model, all three doses prevented cognitive defects, and the highest dose attenuated also disease-related motor phenotypes. In parallel with these behavioral benefits, we detected morphological and functional recovery of synaptic transmission that involved both excitatory and inhibitory synapses on striatal MSNs. Striatal infusion of cholesterol in HD mice also increased levels of the brain-specific cholesterol catabolite 24S-OHC and enhanced endogenous cholesterol biosynthesis, restoring the primary cholesterol defect in HD. At the cellular level, we show that striatal infusion of cholesterol reduced muHTT aggregates by reducing lysosome accumulation.

## Results

### Striatal infusion of cholesterol prevents motor and cognitive deficits in HD mice

To identify the target therapeutic dose of cholesterol to administer, we infused three escalating doses of cholesterol—15 μg (chol-low), 185 μg (chol-medium), and 369 μg (chol-high)—directly into one hemi-striatum of the R6/2 transgenic model of HD (Mangiarini *et al*, 1996). For this purpose, we used miniature infusion osmotic pumps implanted subcutaneously on the back and connected to a catheter. We targeted the striatum as the most affected brain region in HD and the earliest and most obvious site of decreased cholesterol biosynthesis (Shankaran *et al*, 2017). Mice were operated at age 7 weeks, and motor and cognitive tests were performed over a 4-week infusion period (Fig 1A).

Before testing HD mice and to reduce the number of animals in the main study without compromising statistical power, we performed a pilot experiment with healthy wild-type (wt) mice to assess the behavioral influence of osmotic mini-pump implantation and 4 weeks of high-dose cholesterol administration. Mini-pumps filled with artificial cerebrospinal fluid (ACSF) or high-dose cholesterol were implanted in wt mice, and behavioral tests were performed. Coordination, motor activity, and memory recognition were similar among unoperated wt, wt ACSF, and wt chol-high mice (Fig EV1A–F). Using GC-MS, we verified increased cholesterol content in the infused striatum and ipsilateral cortex of wt chol-high mice compared to wt ACSF animals (Fig EV1G and H). These results allowed us to include only unoperated wt mice as controls in subsequent studies.

To visualize the spread of exogenous cholesterol and study its partition in striatum, we tested an analogue of cholesterol tagged with a fluorescent BODIPY group at carbon 24 (BODIPY-chol), using the experimental paradigm described in Fig 1A. After a 4-week infusion period, BODIPY–chol covered 39.7% ± 5.9% of the infused hemi-striatum of R6/2 mice, whereas we found no signal in the contralateral hemisphere (Fig 1B). Within the striatum, BODIPY–cholesterol did not localize with TGN46, a marker of Golgi or with calnexin, a marker of endoplasmic reticulum (ER) (Fig EV1I–J). Instead, BODIPY-cholesterol partially co-localized with late endosome marker Rab9A and with plasma membrane marker PMCA-ATPase (Fig EV1K–L) while complete co-localization was found with LAMP1, a marker of lysosomes (Fig EV1M).

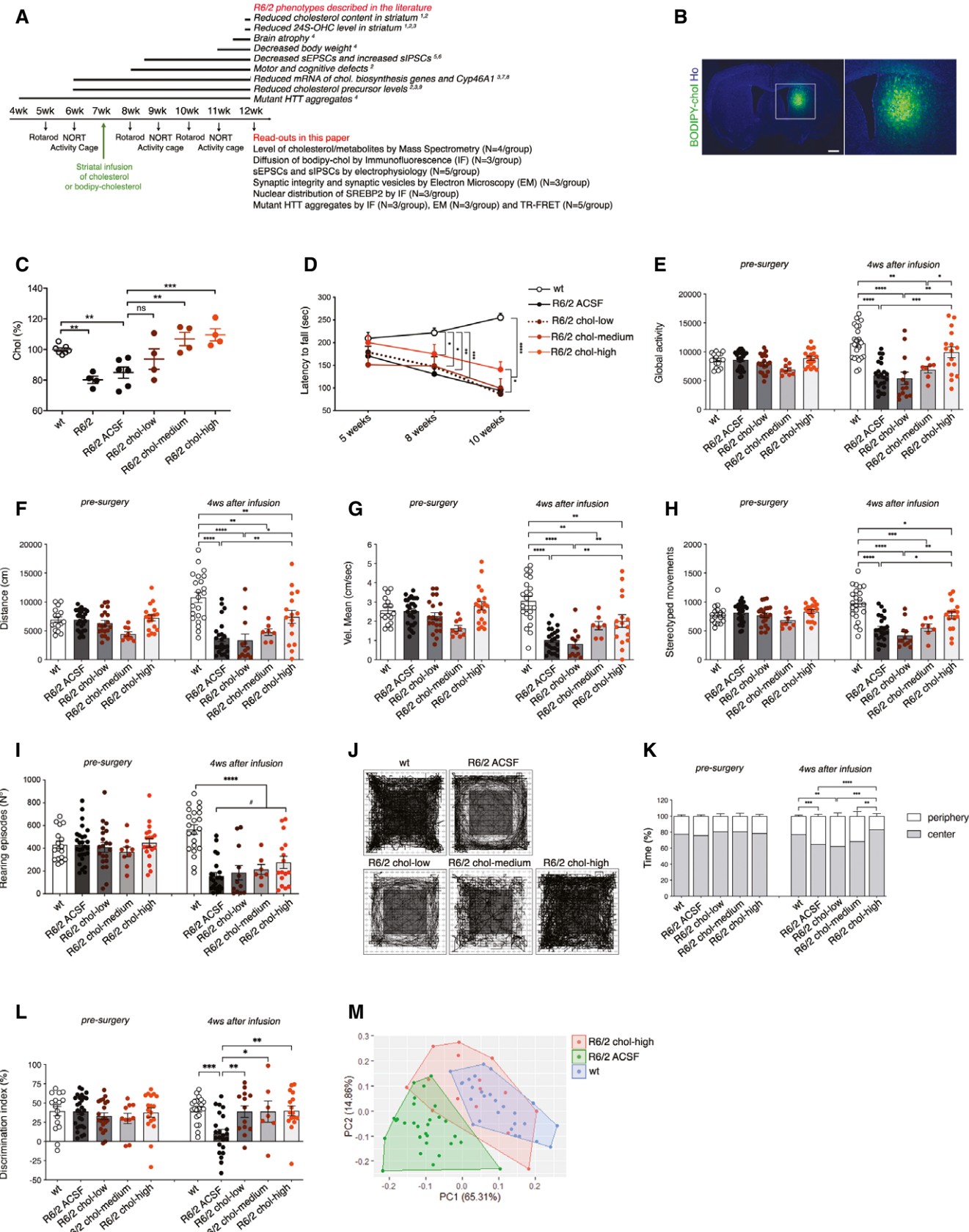

**Figure 1.**

◀

**Figure 1.  Striatal infusion of three escalating doses of cholesterol ameliorates cognitive and motor deficits in HD mice.**

A  Experimental paradigm performed in all the trials: Mini-pumps were implanted in the striatum of 7-week-old mice. A battery of behavioral tests was performed before and after mini-pump implantation. In bold, a list of R6/2 phenotypes (above) and read-outs analyzed following cholesterol (chol) infusion (below). References: (1) Valenza (2010); (2) Valenza et al (2015a,b); (3) Boussicault et al (2016); (4) Mangiarini et al (1996); (5) Cepeda et al (2003); (6) Cepeda et al (2004); (7) Valenza et al (2005); (8) Bobrowska et al (2012); (9) Valenza et al (2007a,b).

B  Representative large image of a brain slice from R6/2 mice after 4-week striatal infusion of fluorescent cholesterol (BODIPY-chol, green). In the inlet, infused striatum is shown. Hoechst (Ho, blue) was used to counterstain nuclei. Scale bar: 1000 μm.

C  Cholesterol content measured by mass spectrometry in infused striatum of untreated wt (N = 7) and R6/2 (N = 4), R6/2 ACSF (N = 5), R6/2 chol-low (N = 5), R6/2 chol-medium (N = 5), and R6/2 chol-high (N = 5) mice at 12 weeks of age after 4-week striatal cholesterol infusion. The low, medium, and high doses, respectively, correspond to 15 μg, 185 μg, and 369 μg of cholesterol infused in the striatum after 4 weeks.

D  Latency to fall (seconds) from an accelerating rotarod from 5 weeks (pre-surgery; i.e., before pump implantation) to 10 weeks of age in wt (N = 23 at 5 weeks, N = 28 at 8 and 10 weeks), R6/2 ACSF (N = 35 at 5 weeks, N = 31 at 8 weeks and N = 30 at 10 weeks), R6/2 chol-low (N = 22 at 5 weeks, N = 14 at 8 weeks and N = 13), R6/2 chol-medium (N = 13 at 5 weeks, N = 9 at 8 weeks and N = 8 at 10 weeks), and R6/2 chol-high (N = 19 at 5 weeks, N = 17 at 8 and 10 weeks) mice. The graph shows means ± standard error for each time point.

E–I  Global motor activity (E), total distance traveled (F), mean velocity (G), stereotyped movements (H), and number of rearings (I) in an open-field test at 6 weeks of age (pre-surgery) and 11 weeks of age (4 weeks after infusion) of wt (N = 16 at 9 weeks, N = 22 at 11 weeks), R6/2 ACSF (N = 27 at 9 weeks, N = 23 at 11 weeks), R6/2 chol-low (N = 20 at 9 weeks, N = 13 at 11 weeks), R6/2 chol-medium (N = 9 at 9 weeks, N = 7 at 11 weeks), and R6/2 chol-high (N = 18 at 9 weeks, N = 16 at 11 weeks) mice.

J, K  Representative track plots (J) generated from the open-field test from wt (N = 16 at 9 weeks, N = 22 at 11 weeks), R6/2 ACSF (N = 26 at 9 weeks, N = 23 at 11 weeks), R6/2 chol-low (N = 11 at 9 weeks, N = 7 at 11 weeks), R6/2 chol-medium (N = 11 at 9 weeks, N = 7 at 11 weeks), and R6/2 chol-high mice (N = 18 at 9 weeks, N = 16 at 11 weeks) and relative quantification (K) of the time spent in the center and in the periphery (%) of the arena.

L  Discrimination index (DI; %) in the novel object recognition test at 6 weeks of age (before pump implantation) and 11 weeks of age (4 weeks after cholesterol infusion) of wt (N = 16 at 9 weeks, N = 21 at 11 weeks), R6/2 ACSF (N = 28 at 9 weeks, N = 23 at 11 weeks), R6/2 chol-low (N = 21 at 9 weeks, N = 13 at 11 weeks), R6/2 chol-medium (N = 10 at 9 weeks, N = 7 at 11 weeks), and R6/2 chol-high mice (N = 18 at 9 weeks, N = 16 at 11 weeks). DI above zero indicates a preference for the novel object; DI below zero indicates a preference for the familiar object.

M  Principal component analysis by combining all the values related to motor and cognitive tasks from wt mice (blue dots), R6/2 ACSF mice (green dots), and R6/2 chol-high mice (red dots).

Data information: The data in (C–L) are from five independent trials and shown as scatterplot graphs with means ± standard error. Each dot (C, E–I, L) corresponds to the value obtained from each animal. Statistics: one-way ANOVA with Newman–Keuls post hoc test (*P < 0.05; **P < 0.01; ***P < 0.001; ****P < 0.0001).

In all trials, with the aim of obtaining the maximum solubility and diffusion of the exogenous cholesterol (Loftsson et al, 2005), we used water-soluble methyl-β-cyclodextrin (MβCD)-balanced cholesterol. To exclude any potential effect of MβCD, we performed motor and cognitive tests in an additional control group, comparing R6/2 ACSF and R6/2 mice with mini-pumps containing ACSF and the equivalent quantity of MβCD to be used with chol-high (R6/2 ACSF-MβCD). We found that the presence of MβCD in ACSF did not influence outcomes in the motor and cognitive tasks (Fig EV2A–F). Furthermore, cholesterol content and levels of cholesterol precursors and 24S-OHC were not influenced by MβCD (Fig EV2G–K).

We next tested three doses of exogenous cholesterol in R6/2 mice. We used GC-MS to quantify cholesterol content in the striatum and cortex of R6/2 mice and verify the success of chronic cholesterol infusion. Compared to animals implanted with osmotic mini-pumps filled with ACSF, R6/2 mice infused with the three doses of cholesterol showed a dose-dependent increase in cholesterol content in the striatum (Fig 1C). The increase was significant with the chol-medium and chol-high doses (Fig 1C). Of note, striatal cholesterol content was consistently decreased in late-symptomatic R6/2 and R6/2 ACSF mice (for a total of 20 MS measurements performed in 10 mice) compared to wt mice (14 MS measurements in 7 mice; Figs 1C and EV3B; Valenza et al, 2010). A significant increase in cholesterol content was also observed in the ipsilateral cortex of R6/2 chol-high groups (Fig EV3A) but not in the contralateral striatum and cortex of these animals compared to R6/2 ACSF mice (Fig EV3B and C). These results demonstrate the efficiency of osmotic mini-pumps in releasing exogenous cholesterol around the site of infusion and partially into the surrounding cortex in HD animals.

For all the experiments, the body weight of mice was monitored over time. Before the surgery, at 7 weeks of age, body weight was similar across all groups (Fig EV3D and E). At 12 weeks of age, R6/2-ACSF males, as well as R6/2 chol-low and R6/2 chol-high males, lost body weight compared to wt males. Of note, R6/2 chol-high males did not lose body weight over time as they were comparable to that of wt mice (Fig EV3F). No significant changes were observed in all female groups regardless of genotype or treatment (Fig EV3G).

Compared to wt mice, R6/2 ACSF animals showed a progressive deterioration in fine motor coordination, as assessed by an accelerating rotarod test, from the early-symptomatic (8 weeks of age) to late-symptomatic stages (10 weeks of age; Fig 1D). In contrast, R6/2 chol-high mice presented a partial but significant amelioration in rotarod performance at 10 weeks compared to R6/2 chol-medium and R6/2 chol-low groups (Fig 1D).

To further test motor abilities, we evaluated spontaneous locomotor activity in the activity cage test. R6/2 ACSF mice exhibited a severe hypokinetic phenotype with disease progression, i.e., at 11 weeks (Fig 1E, right panel) compared to 6 weeks (Fig 1E, left panel; Fig 1E). Of note, global activity deficits were normalized in R6/2 chol-high mice, while the low and medium dose of cholesterol did not produce any effect (Fig 1E). Other parameters, such as distance traveled (Fig 1F), mean velocity (Fig 1G), and stereotyped movements (Fig 1H), significantly improved in R6/2 chol-high mice compared to R6/2 ACSF mice. An ANOVA multiple comparison test revealed a significant decrease in the number of vertical movements (rearings) in all R6/2 groups (Fig 1I). However, a significant increase in rearings was found only in R6/2 chol-high group (P = 0.0419, unpaired t-test), indicating an effect of exogenous cholesterol on this parameter only at the highest dose. By comparing among R6/2 groups treated with the three doses of cholesterol, we identified a dose-dependent effect, with a progressive increase in

all activity-related values from low to medium to high doses of cholesterol (Table EV1).

As a measure of anxiety-like behavior, we also evaluated the time that mice spent exploring the periphery or center area of the arena during the activity cage test (Fig 1J). R6/2 ACSF animals spent more time in the periphery compared to wt mice, indicating anxiety-related behavior. R6/2 chol-high mice spent more time in the center compared to R6/2 ACSF mice, with animals performing similarly to the wt group (Fig 1K), indicating a normalization of anxiety-related behavior.

To assess if striatal infusion of cholesterol influences also cognitive abilities, we used the novel object recognition test (NORT). Long-term memory declined during disease progression in R6/2 ACSF mice, with a marked impairment in the ability to discriminate novel and familiar objects at age 11 weeks (Fig 1L). R6/2 mice in all cholesterol-dose groups performed similarly to wt mice (Fig 1L). Finally, principal component analysis (PCA) of all values related to motor and cognitive tests for chol-high animals identified two distinguishable groups (wt and R6/2 ACSF) that separated in the first principal component, with the R6/2 chol-high mice displaying a greater overlap with wt group than R6/2 ACSF mice (Fig 1M).

Taken together, these results indicate that extensive behavioral recovery occurs in HD mice after striatal infusion of cholesterol.

## Striatal infusion of cholesterol rescues excitatory synaptic defects in HD mice

Cholesterol is involved in synaptic function (Pfrieger, 2003), and functionality and morphology of excitatory and inhibitory synapses are both altered in HD (Cepeda et al, 2003, 2004). For this reason, we adopted a combination of techniques to explore whether and how exogenous cholesterol can influence on synaptic function and morphology. The analyses were performed in R6/2 chol-high mice and relevant controls.

We first compared whole-cell patch-clamp recordings of striatal MSNs from brain slices of wt, R6/2 ACSF, and R6/2 chol-high mice (Fig 2A). The membrane capacitance, which is proportional to cell size, was significantly lower in R6/2 ACSF compared to wt MSNs and unaffected in R6/2 chol-high mice (Table EV2). Similarly, input resistance, reflecting the number of ion channels expressed by the cell, was significantly increased in both R6/2 ACSF and R6/2 chol-high compared to wt cells, but unaffected by cholesterol administration (Table EV2).

To evaluate the effect of cholesterol on excitatory transmission, we recorded spontaneous excitatory postsynaptic currents (sEPSC) in MSNs at a holding potential of −70 mV (Fig 2B). We did not find any significant differences in the average amplitude of sEPSCs between groups (Fig 2C). In agreement with previous studies (Cepeda et al, 2003), we found a significant reduction in frequency of sEPSCs in R6/2 MSNs compared to wt MSNs (Fig 2D). Of note, striatal infusion of cholesterol led to a significant increase in the frequency of sEPSCs in R6/2 chol-high compared to R6/2 ACSF mice, partially rescuing this defect (Fig 2D).

To identify the structural bases underlying the functional recovery of excitatory synapses after striatal infusion of cholesterol, we undertook a series of morphological studies by electron microscopy. We employed the combination of focused ion beam milling and scanning electron microscopy (FIB/SEM) followed by the 3D reconstruction of complete synaptic junctions in large volumes of tissue (Fig 2E and F). The high spatial resolution of the FIB/SEM images and long series of serial sections allowed for classification of all synapses as asymmetric or symmetric using morphological criteria (Merchán-Pérez et al, 2009), providing the actual number of synapses per volume of the striatal region. Fig 2F shows an example of the 3D reconstruction of excitatory synapses (in yellow) in a large portion of the tissue blocks used for the analysis (10 μm × 5 μm × 10 μm) from wt, R6/2, R6/2 ACSF, and R6/2 chol-high mice. The density of excitatory synapses was reduced in the striatal neurons of R6/2 compared to wt mice, but cholesterol infusion did not rescue this defect (Fig 2G). We then tested whether cholesterol could influence synaptic parameters at the active site of excitatory synapses. Using transmission electron microscopy (TEM), we visualized the synaptic vesicles (SVs) to quantify their density (Fig 2H and I). The number of total and docked SVs was reduced in R6/2 and R6/2 ACSF mice compared to wt mice (Fig 2J and K). These structural defects were completely rescued by striatal infusion of cholesterol in R6/2 chol-high mice (Fig 2J and K). Instead, postsynaptic density (PSD) area and length (Fig EV4A and B), pre-synaptic area, and active zone (AZ) length (Fig EV4C and D) were not altered in R6/2 groups compared to wt mice.

Collectively, these findings indicate that cholesterol partially rescues excitatory synaptic transmission by enhancing the formation and/or release of SVs at the pre-synaptic site, but not by increasing the number of excitatory synapses.

## Striatal infusion of cholesterol rescues GABAergic inhibitory synaptic defects in HD mice

To test the effect of exogenous cholesterol at the inhibitory synapses, we recorded spontaneous inhibitory synaptic currents (sIPSCs) in brain slices from wt, R6/2 ACSF, and R6/2 chol-high mice at a holding potential of 0 mV (Fig 3A). The average amplitude of sIPSCs was similar between wt and R6/2 ACSF MSNs and was unaffected by cholesterol (Fig 3B). However, the average frequency of sIPSCs was significantly increased in R6/2 ACSF compared to wt cells (Fig 3C) as known in the literature (Cepeda et al, 2004). Of note, striatal infusion of cholesterol led to a significant reduction in the average frequency of sIPSCs, bringing this parameter close to what we observed in wt MSNs (Fig 3C) and indicating that exogenous cholesterol contributes to restoring GABAergic inhibitory synaptic defects.

To identify the structural changes underlying the functional recovery of inhibitory synaptic transmission after striatal infusion of cholesterol, we first analyzed the number of inhibitory synapses per volume of striatal tissue, looking at the serial sections obtained by FIB/SEM. We identified symmetric junctions by the presence of a thin postsynaptic density and performed 3D reconstruction for all groups (Fig 3D). The density of inhibitory synapses was significantly increased in striatal neurons in both R6/2 and R6/2 ACSF mice compared to wt mice (Fig 3E), in agreement with the electrophysiological findings of increased frequency. Cholesterol striatal infusion reduced the density of inhibitory synapses, rescuing the morphological defect (Fig 3E). TEM analysis of inhibitory synapses showed no alterations in SVs density in R6/2 and R6/2 chol-high mice compared to wt animals (Fig EV4E and F).

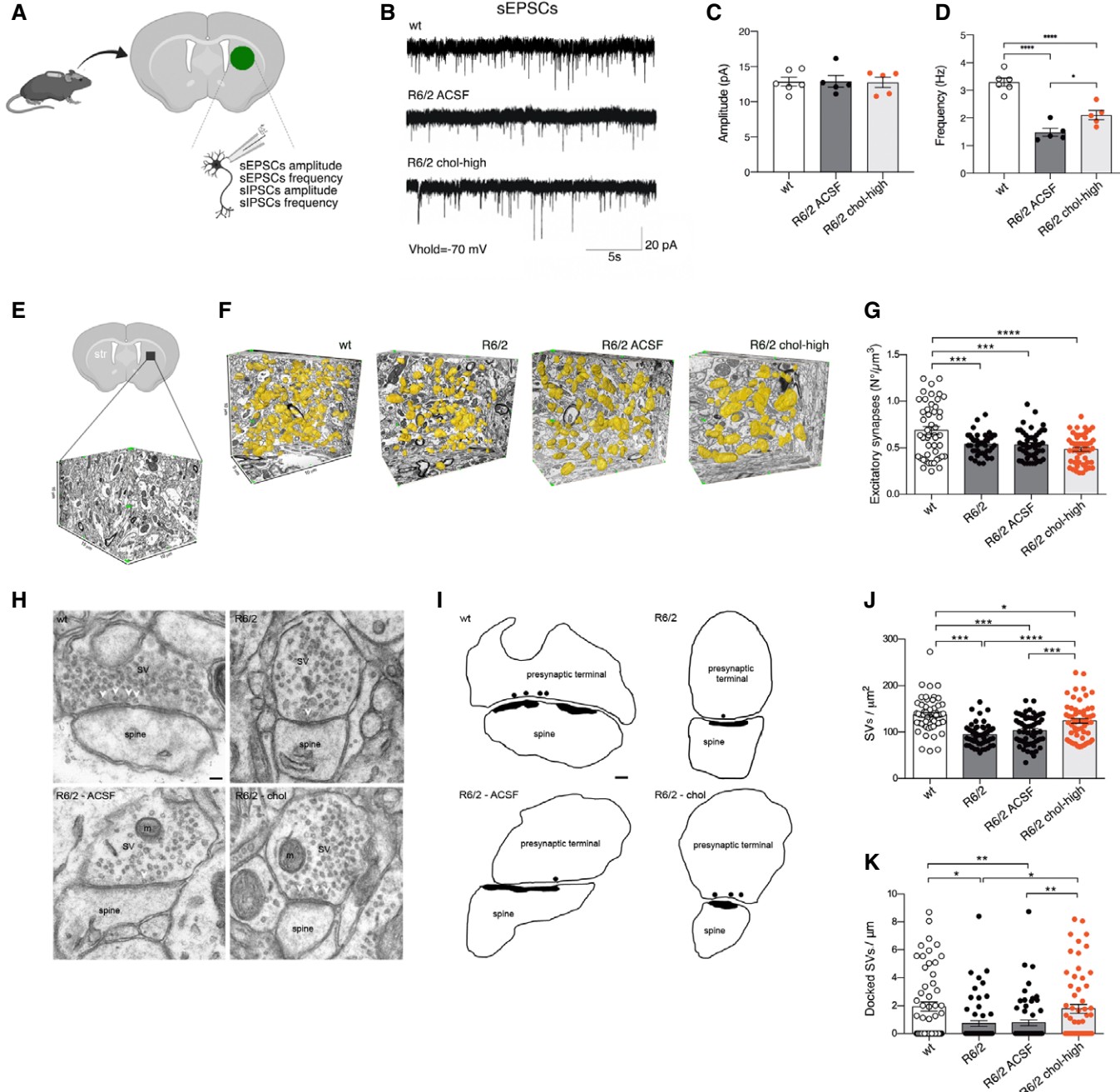

**Figure 2. Striatal infusion of the high dose of cholesterol partially rescues synaptic activity and ultrastructure of excitatory synapses in HD MSNs.**

A   Schematic representation of the electrophysiological parameters analyzed in the infused striatum of 12-week-old mice following 4 weeks of striatal infusion of cholesterol.

B   Spontaneous EPSCs were recorded from striatal MSNs (wt = 6; R6/2 ACSF = 5; R6/2 chol-high = 5) at a holding potential of −70 mV.

C, D   Average amplitude (C) and average frequency (D) of EPSCs from wt, R6/2 ACSF, and R6/2 chol-high MSNs.

E–G   Number of excitatory synapses per volume of striatum by using FIB/SEM followed by 3D reconstruction. (E) Representative tissue block of striatum (10 μm × 15 μm × 10 μm) used for the 3D analysis. (F) Representative FIB-SEM segmentation and reconstruction of excitatory synapses (yellow) in 200 serial sections of striatum for a total volume of 500 μm³, of wt, R6/2, R6/2 ACSF, and R6/2 chol-high mice. (G) Density of excitatory synapses in at least 1,500 μm³ of striatal tissue from wt, R6/2, R6/2 ACSF, and R6/2 chol-high mice (N = 3 animals/group).

H, I   TEM images (H) and 2D profile (I) of pre-synaptic terminal sections contain SVs and mitochondria (m). Docked vesicles are indicated by a white arrowhead (H) and black circle (I) and are defined as a vesicle with its center located within 20 nm from the pre-synaptic membrane. Scale bar: 100 nm.

J, K   Quantification showing the total SVs/μm² (J) and the docked SVs/μm (K) in R6/2, R6/2 ACSF, and wt striatal synapses (N = 3 animals/group). N = 60 excitatory synapses/group were counted.

Data information: The data in (C, D, G, J, and K) are shown as scatterplot graphs with means ± standard error. Each dot corresponds to the value of each cell recorded (C and D), the number of synapses counted/μm³ in different blocks of striatal tissue (G), and the number of total SVs/μm² (J) and docked SVs/μm of active zone (K) counted in each synapse. Statistics: one-way ANOVA with Newman–Keuls *post hoc* test (*P < 0.05; **P < 0.01; ***P < 0.001; ****P < 0.0001).

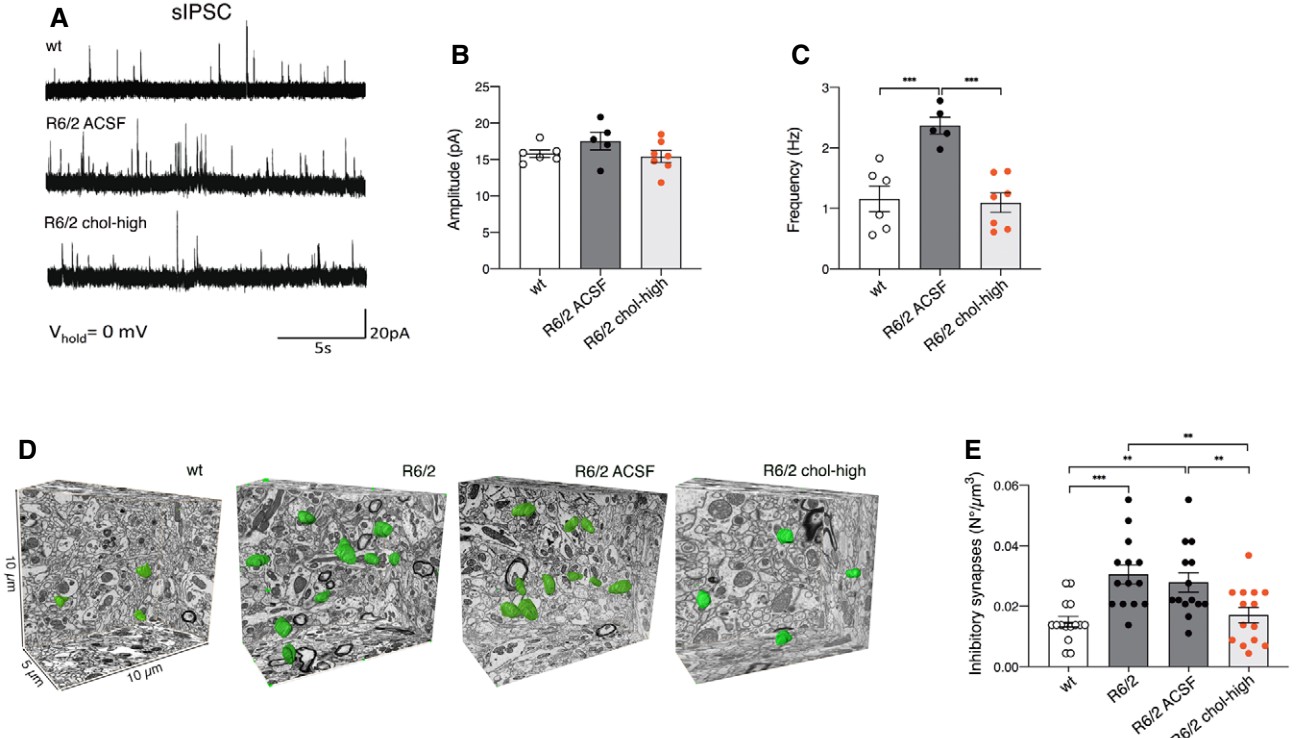

**Figure 3. Striatal infusion of the high dose of cholesterol rescues synaptic activity and ultrastructure of inhibitory synapses in MSNs of HD mice.**

A Spontaneous IPSCs were recorded from striatal MSNs (wt = 6; R6/2 ACSF = 5; R6/2 chol-high = 5) at a holding potential of 0 mV.

B, C Average amplitude (B) and average frequency (C) of IPSCs from wt, R6/2 ACSF, and R6/2 chol-high MSNs.

D, E Evaluation of the number of inhibitory synapses per volume of striatum by using FIB/SEM followed by 3D reconstruction. (D) Representative FIB-SEM segmentation and reconstruction of inhibitory synapses (green) in 200 serial sections of striatum for a total volume of 500 $\mu m^3$ in wt, R6/2, R6/2 ACSF, and R6/2 chol-high mice. (E) Density of inhibitory synapses in at least 2000 $\mu m^3$ of striatal tissue of wt, R6/2, R6/2 ACSF, and R6/2 chol-high mice (N = 3 mice/group).

Data information: The data in (B, C, and E) are shown as scatterplot graphs with means ± standard error. Each dot corresponds to the value of each cell recorded (B, C) and to the number of synapses counted/$\mu m^3$ in different blocks of tissue for each group of animals (N = 3 animals/group) (E). Statistics: one-way ANOVA with Newman–Keuls *post hoc* test (**$P < 0.01$; ***$P < 0.001$).

These findings indicate that striatal infusion of cholesterol acts differentially on excitatory and inhibitory synapses and rescues alterations in inhibitory synaptic transmission by reducing the number of inhibitory synapses.

## Striatal infusion of cholesterol does not rescue myelin defects in HD mice

Cholesterol influences myelin membrane biogenesis and the functionality of mature myelin (Saher & Stumpf, 2015). To evaluate whether striatal infusion of cholesterol counteracts myelin deficits in HD mice, we examined myelin in the striatum and corpus callosum of wt, R6/2, R6/2 ACSF, and R6/2 chol-high mice at age 12 weeks. The G-ratio of myelinated axons, a measure of myelin sheath thickness as evaluated by electron microscopy, was increased in both the striatum and corpus callosum of all R6/2 groups compared to wt mice (Appendix Fig S1A–F), indicating thinner myelin sheaths in HD mice even after striatal infusion of cholesterol. Periodicity, a measure of myelin compaction calculated as the mean distance between two major dense lines was similar in the striatum and in the corpus callosum among all groups (Appendix Fig S1G–I). These data suggest the presence of a thinner myelin sheath in the striatum

and corpus callosum of R6/2 mice and that cholesterol treatment did not rescue this defect.

## Striatal infusion of cholesterol induces changes in sterol metabolism in HD mice

The synthesis of new cholesterol and production of its neuronal-specific catabolite 24S-OHC are closely related (Lund *et al*, 2003). To maintain constant levels of cholesterol in the brain, any excess of cholesterol is catabolized into 24S-OHC that can cross the blood–brain barrier and enter the circulation (Björkhem *et al*, 1997; Leoni *et al*, 2008, 2013).

Figure 4A shows a schematic representation of the enzymes involved in cholesterol biosynthesis and catabolism and how they are affected in HD. In this study, we first measured 24S-OHC level by ID-MS and found reductions in the contralateral and ipsilateral striatum of R6/2 ACSF compared to wt mice (Fig 4B). 24S-OHC level was increased in the infused striatum of R6/2 chol-high mice compared to R6/2 ACSF mice, with higher levels compared to wt mice (Fig 4B). Student's *t*-test analysis revealed a significant increase in 24S-OHC level in the infused striatum of wt mice treated with the high dose of cholesterol compared to wt ACSF mice,

suggesting a genotype-independent effect on 24S-OHC level (Fig EV5A). The low dose of cholesterol did not affect striatal level of 24S-OHC in either wt or R6/2 mice (Fig EV5A and B).

Exogenous cholesterol might operate in negative feedback on endogenous cholesterol biosynthesis, which is already compromised in HD mice. A robust deficit in levels of the key cholesterol precursors lanosterol and lathosterol was found in the striatum of R6/2 and R6/2 ACSF mice compared to wt animals (Fig 4C and D), confirming previous results (Valenza *et al*, 2007b, 2010). Unexpectedly, we also found a significant increase in striatal levels of lanosterol, lathosterol, and desmosterol in R6/2 chol-high mice compared R6/2 or R6/2 ACSF animals (Fig 4C–E), indicating enhancement of endogenous cholesterol biosynthesis following striatal cholesterol infusion. This increase was specific for the infused striatum and was not observed in the contralateral striatum of the same mice (Fig 4C–E). Increased levels of all cholesterol precursors were also found in wt mice treated with the high dose of cholesterol compared to wt ACSF mice (Fig EV5C, E, G), whereas we observed no changes in wt or R6/2 mice treated with the low dose of cholesterol (Fig EV5C–H).

Translocation into the nucleus of the N-terminal (active) fragment of SREBP2 triggers expression of genes involved in cholesterol biosynthesis (Brown & Goldstein, 1997). We sought to assess whether nuclear translocation of SREBP2 mediates the increase in endogenous cholesterol biosynthesis after striatal cholesterol infusion. For this purpose, we performed immunofluorescence staining with a specific antibody that targets the N-terminal fragment of this protein. As expected, SREBP2 localization was both nuclear and perinuclear in the striatum of wt mice (Fig 4F and G). On the contrary, reduced nuclear distribution of SREBP2 was found in the striatum of untreated R6/2 mice compared to wt mice (Fig 4F and G), as expected on the basis of the evidence of reduced cholesterol synthesis in R6/2 striatum. As shown in Fig 4H, we found a marked increase in nuclear distribution of SREBP2 in the infused striatum compared to the contralateral striatum of R6/2 chol-high mice, as confirmed by the relative quantification (Fig 4I). Specifically, by coupling the antibody against SREBP2 with a neuronal or an astrocytic marker (NeuN and GFAP, respectively), we found that the increased nuclear localization of SREBP2 was specific for glial cells (Fig 4J and K), the major producers of cholesterol in the adult brain (Mauch *et al*, 2001; Camargo *et al*, 2012; Ferris *et al*, 2017).

Taken together, these findings indicate that the high dose of cholesterol enhances 24S-OHC availability and that increased endogenous cholesterol biosynthesis may occur through nuclear translocation of SREBP2 in glial cells.

## Striatal infusion of cholesterol induces clearance of muHTT aggregates in R6/2 mice

A hallmark of HD is the presence of intracellular aggregates of muHTT (DiFiglia *et al*, 1997; Gutekunst *et al*, 1999; Maat-Schieman *et al*, 1999; Herndon *et al*, 2009). To test whether striatal infusion of cholesterol influences muHTT aggregation, we employed different methods to visualize and quantify different forms of muHTT during the process of aggregation. We first performed immunofluorescence staining on brain sections of R6/2 ACSF and R6/2 chol-high mice by using the EM48 antibody, which is specific for the expanded polyQ tract prone to aggregate (Fig 5A). The number and size of EM48-positive aggregates (size aggregates ≈ 2 μm) was similar in the striatum of both hemispheres in R6/2 ACSF mice but significantly reduced in the infused striatum compared to the contralateral striatum in R6/2 chol-high mice (Fig 5B and C; Appendix Fig S2A and B). Evidence of reduced muHTT aggregates in cortical tissues of R6/2 mice, however, was variable among the animals (Appendix Fig S2C and D), likely depending on a heterogeneous diffusion of cholesterol into the cortex. We did not observe fewer aggregates in the hippocampus of the same animals (Appendix Fig S2E and F).

Double immunofluorescence staining with EM48 antibody in combination with an antibody against DARPP32 (MSN marker) or GFAP (astrocyte marker) allowed us to count the number of muHTT nuclear aggregates in the striatum of R6/2 chol-high mice in different cell types. In the infused compared to the contralateral striatum of R6/2 chol-high mice, the number of nuclear aggregates was reduced 6-fold in neurons (Fig 5D and E) and 2-fold in astrocytes (Fig 5F and G).

To investigate the morphology and localization of muHTT aggregates after striatal infusion of cholesterol, we employed electron

---

**Figure 4. Striatal infusion of the high dose of cholesterol increases endogenous cholesterol catabolism and synthesis in the striatum of HD mice.**

A   Schematic pathway of cholesterol synthesis and cholesterol precursors and catabolites in the brain. Green arrows indicate enzymes with downregulated mRNA in HD, and red arrows indicate cholesterol precursors or metabolites that were decreased per ID-MS; SREBP2-dependent genes are in bold. Here, lanosterol, 7-lathosterol, desmosterol, and 24S-OHC (highlighted in the boxes) were measured by ID-MS. References: (1) Bobrowska *et al* (2012); (2) Lee *et al* (2015); (3) Valenza *et al* (2005); (4) Samara *et al* (2014); (5) Boussicault *et al* (2016); (6) Kacher *et al* (2019); (7) Valenza *et al* (2007a); (8) Valenza *et al* (2007b); (9) Shankaran *et al* (2017); (10) Valenza *et al*, 2010; (11) Valenza *et al* (2015a,b).

B   24S-OHC level measured by mass spectrometry in the infused and contralateral striatum of wt, R6/2-ACSF, and R6/2 chol-high mice at 12 weeks of age after a 4-week striatal infusion of cholesterol (*N* = 4/group). All values are expressed as % above the mean of wt, and these data are shown as scatterplots with means ± standard error. Each dot corresponds to the value obtained from each animal.

C–E   Lanosterol (C), lathosterol (D), and desmosterol (E) level measured by mass spectrometry in the infused and contralateral striatum of wt, R6/2 ACSF, and R6/2 chol-high mice at 12 weeks of age after 4-week striatal infusion of cholesterol (*N* = 4/group). All values are expressed as % above the mean of wt, and these data are shown as scatterplots with means ± standard error. Each dot corresponds to the value obtained from each animal.

F–I   Nuclear (white arrows) and perinuclear (yellow arrows) localization of endogenous SREBP2 in the striatum of wt, R6/2, and R6/2-chol mice. Representative confocal image (F and H) and relative quantification (G and I) of SREBP2 (red) in wt and R6/2 mice and in the infused and contralateral striatum of R6/2 chol-high mice (*N* = 4). Hoechst (Ho, blue) was used to counterstain nuclei. Scale bar in F and H: 10 μm. Graphs in (G and I) represent the intensity of SREBP2 normalized on nuclei (%). Data are shown as scatterplots with means ± standard error. Each dot corresponds to the value obtained from each image. Statistics: Student's *t*-test (**$P < 0.01$; ****$P < 0.0001$).

J, K   Representative high-magnification confocal images of immunostaining against SREBP2 (red) and NeuN (J) or GFAP (K) (green) on coronal sections of brains from R6/2 chol-high mice. A cell in the infused striatum, positive for NeuN and GFAP, respectively, is shown. Hoechst (Ho, blue) was used to counterstain nuclei. Scale bars: 1 μm.

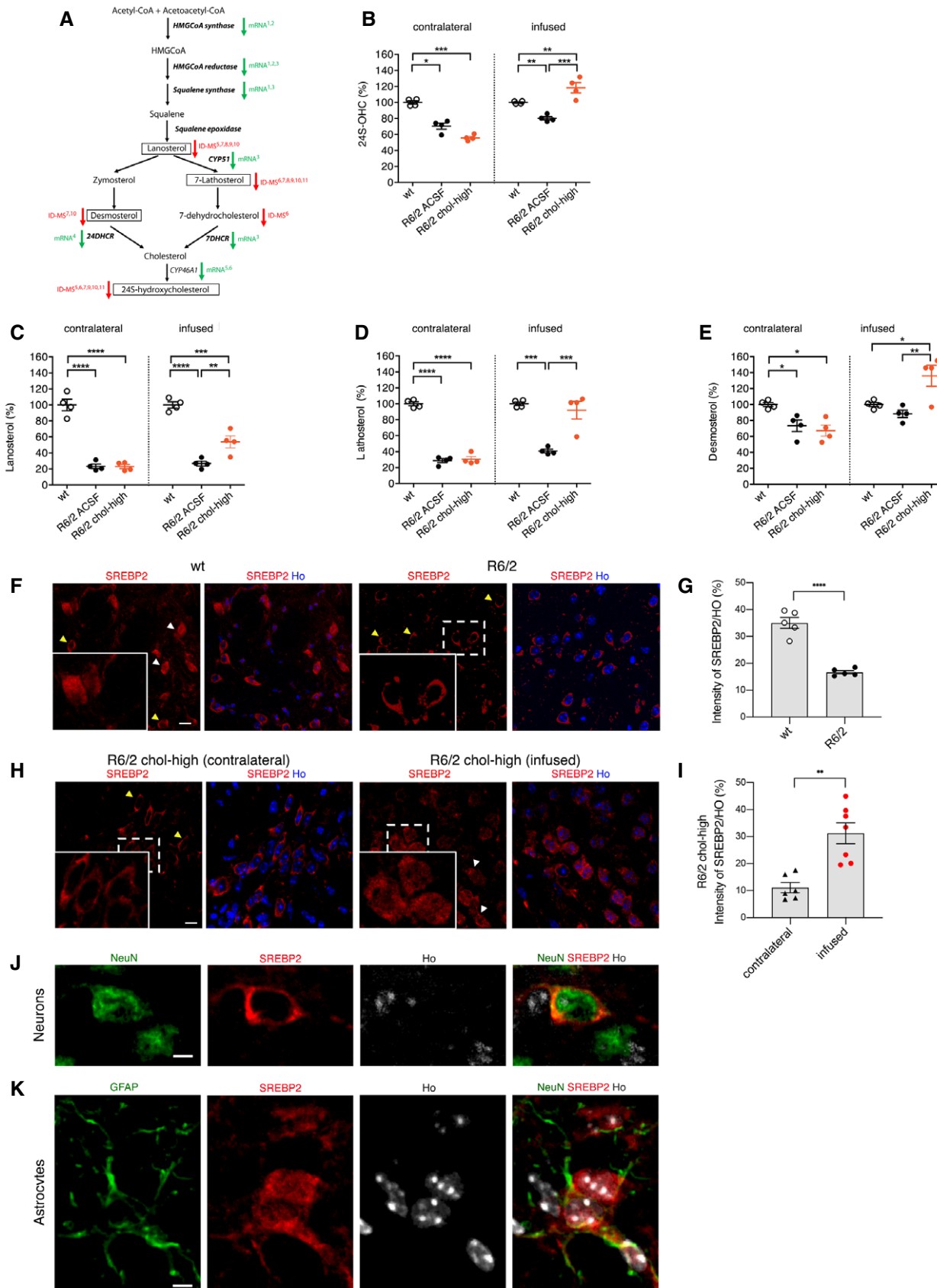

Figure 4.

microscopy using pre-embedded immunogold labeling for EM48 to visualize muHTT in striatal neurons of wt, R6/2, R6/2 ACSF, and R6/2 chol-high mice. Immunogold-labeled HTT fragments were found either as protofibril-like structures of about 300 nm or dispersed in the cytoplasm and nucleus of striatal neurons from R6/2 ACSF mice (Fig 5H). In contrast, muHTT was found dispersed and

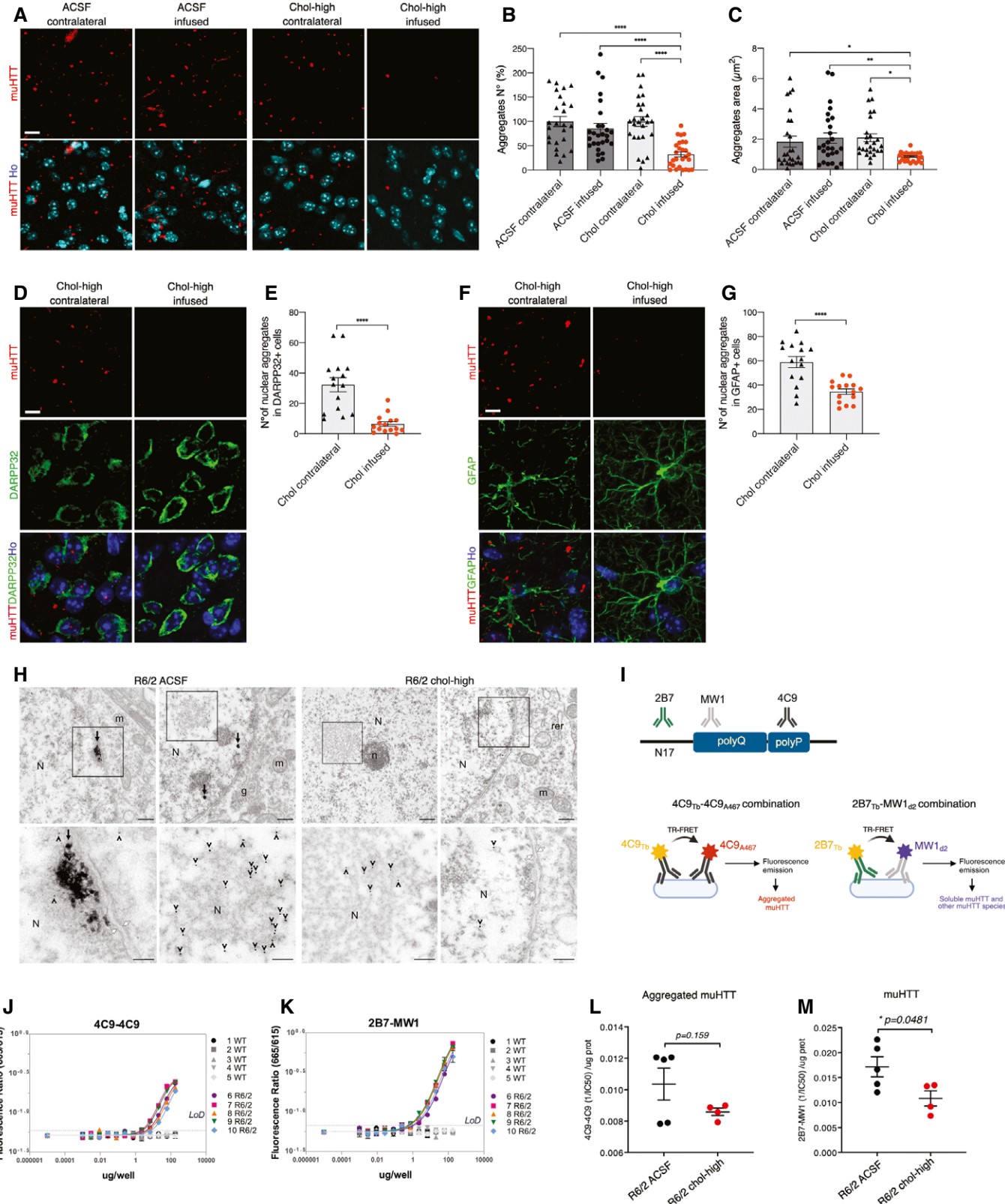

Figure 5.

**Figure 5. Striatal infusion of the high dose of cholesterol rescues muHTT aggregation in the striatum of HD mice.**

A–G Immunolabeling of muHTT aggregates (red) in R6/2 ACSF and R6/2 chol-high mice (N = 3/group). Zoom of representative confocal images of immunostaining against muHTT aggregates (red) showing muHTT aggregates positive for EM48 antibody in the infused and contralateral striatum (A) and relative quantification of number (B) and size (C) of aggregates. Hoechst (Ho, blue) was used to counterstain nuclei. 18 images/animal were analyzed from 9 sections throughout the entire striatum. Representative confocal images of immunostaining against muHTT (red) and DARPP32 (D) or GFAP (F) (green) showing muHTT aggregates positive for EM48 antibody in the infused striatum in neurons or astrocytes and relative quantification (E, G). Hoechst (Ho, blue) was used to counterstain nuclei. All values are expressed as % above the mean of aggregates in the contralateral striatum of R6/2 ACSF or of R6/2 chol-high. The data in (B, C, E, and G) are shown as scatterplots with means ± standard error. Each dot corresponds to aggregates counted in all the images from 3 animals. Scale bars: 10 μm (A) and 5 μm (D, F). Statistics: one-way ANOVA followed by Newman–Keuls multiple comparison tests (*$P < 0.05$; **$P < 0.01$; ****$P < 0.0001$).

H TEM images of EM48 pre-embedding immunogold labeling showing muHTT aggregates in the striatal neuron cell bodies of R6/2 ACSF mice and R6/2 chol-high mice (upper panels). In the lower panels, muHTT aggregates are clearly visible and the magnifications show the area indicated by the black square in the upper images. Arrows indicate large muHTT aggregates with a fibrous structure in the nucleus, and arrowheads indicate single 10-nm gold particles in MSN. Nucleus (N), nucleolus (n), mitochondrion (m), rough endoplasmic reticulum (rer), Golgi apparatus (g), and white arrows indicate the nuclear envelope (N = 3 animals/group). Scale bars: 700 nm and 300 nm.

I–M Quantification of aggregated and total muHTT in the infused striatum of HD mice after 4-week cholesterol infusion by TR-FRET analysis using different antibody pairs. Schematic representation of employed TR-FRET assay (I). Preliminary assessment of the sustainability of the assay in wt and R6/2 striata (N = 5/group) using 4C9-4C9 and 2B7-MW1 antibodies in combination to detect, respectively, muHTT aggregates (J), and total muHTT (K). Quantification of muHTT aggregates (L) and soluble and other muHTT species (M) in the infused striata of R6/2 ACSF and R6/2 chol-high mice. Data in (J–M) are shown as scatterplots with means ± standard error. Each dot corresponds to the value obtained from one striatum. Statistics: Student's t-test (*$P < 0.05$).

never composed in a fibril network in striatal neurons from R6/2 chol-high animals (Fig 5H).

We next sought to apply a more reliable quantitative measure of muHTT oligomers during the early phases of aggregation process (Baldo et al, 2012; Weiss et al, 2012). For this purpose, we employed a time-resolved Förster resonance energy transfer (TR-FRET)-based immunoassay (Baldo et al, 2012) to quantify muHTT species in striatal tissues from R6/2 ACSF and R6/2 chol-high mice using specific antibodies. In particular, we used the 4C9-4C9 combination to detect specifically muHTT aggregates, while the 2B7-MW1 combination recognized soluble muHTT and other muHTT species (Fig 5I). First, we validated the feasibility and specificity of the assay in striatal samples of wt and R6/2 mice (Fig 5J and K). Then, we quantified TR-FRET detection of aggregated muHTT (4C9-4C9 combination) and found no difference between R6/2 chol-high and R6/2 ACSF mice (Fig 5L). In contrast, a significant decrease in soluble and other muHTT species (2B7-MW1 combination) was found in the striatum of R6/2 chol-high compared to R6/2 ACSF mice (Fig 5M). Collectively, these findings demonstrate that striatal infusion of cholesterol counteracts the aggregation of different muHTT species which may ultimately contribute to reduced toxicity in HD mice.

**Striatal infusion of cholesterol reverses lysosomal accumulation in HD mice**

We next sought to test whether striatal infusion of cholesterol can stimulate clearance pathways involving autophagy or lysosomal activity. For this purpose, we performed immunofluorescence staining on brain sections of wt, R6/2 ACSF, and R6/2 chol-high mice with antibodies against p62, a protein involved in the recognition and delivery of substrates to autophagosomes, and against the lysosomal-associated membrane protein LAMP1. p62 (red signal) was present in round bodies in the perinuclear area of wt cells but was present primarily as cellular dots in the contralateral and infused striatum of R6/2 ACSF and R6/2 chol-high mice (Fig 6A). Quantification analysis revealed an increase in p62 dots in all R6/2 animal groups compared to wt mice (Fig 6B), suggesting a high basal autophagy in the presence of muHTT, which cholesterol treatment did not significantly influence.

In contrast, immunofluorescence staining for LAMP1 (Fig 6C) and relative quantification (Fig 6D) showed increased LAMP1 density in

the striatum of R6/2 ACSF mice and in the contralateral striatum of R6/2 chol-high compared to wt mice, suggesting an accumulation of lysosomes in HD cells. Of note, LAMP1 density was restored to physiological levels in the infused striatum of R6/2 chol-high mice (Fig 6C).

Taken together, these results suggest that striatal infusion of cholesterol in HD mice may affect lysosome function and counteract muHTT aggregates by reducing their accumulation in HD cells.

# Discussion

In this work, we identified the therapeutic dose of cholesterol that can prevent both motor and cognitive defects in HD mice and ameliorate synaptic transmission while reducing muHTT aggregate load in the brain. Moreover, we showed that all tested doses of infused cholesterol prevented cognitive decline. In particular, the lower dose of cholesterol used here (15 μg) is similar to that employed in our previous work in which we delivered cholesterol via brain-targeted polymeric nanoparticles (Valenza et al, 2015b). The success in preventing mouse cognitive decline in both studies using this cholesterol dose is in line with reports highlighting a link between cholesterol and cognitive impairments in adult patients (Martin et al, 2014; Segatto et al, 2014). In contrast, only the highest dose of cholesterol we used here could also counteract progression of motor defects, suggesting that restoration of motor circuit function may require a higher cholesterol dose.

Cholesterol infusion can exert these beneficial effects by several mechanisms. One is improving the function of residual synaptic circuits. In fact, we showed here that striatal infusion of cholesterol restored both glutamatergic activation and GABAergic inhibition in MSNs of HD mice. Accordingly, exogenous cholesterol increased the number of total and docked vesicles of glutamatergic synapses, accounting for the increasing probability of vesicle release demonstrated through partial normalization of glutamatergic spontaneous synaptic current frequency. Furthermore, exogenous cholesterol reduced the number of GABAergic inhibitory synapses, as also demonstrated by a reduced frequency of spontaneous inhibitory currents. The magnitude of rescue was more evident for inhibitory transmission/synapses, suggesting circuit-specific signaling mechanisms in which cholesterol may act differently. Specific proteins involved in

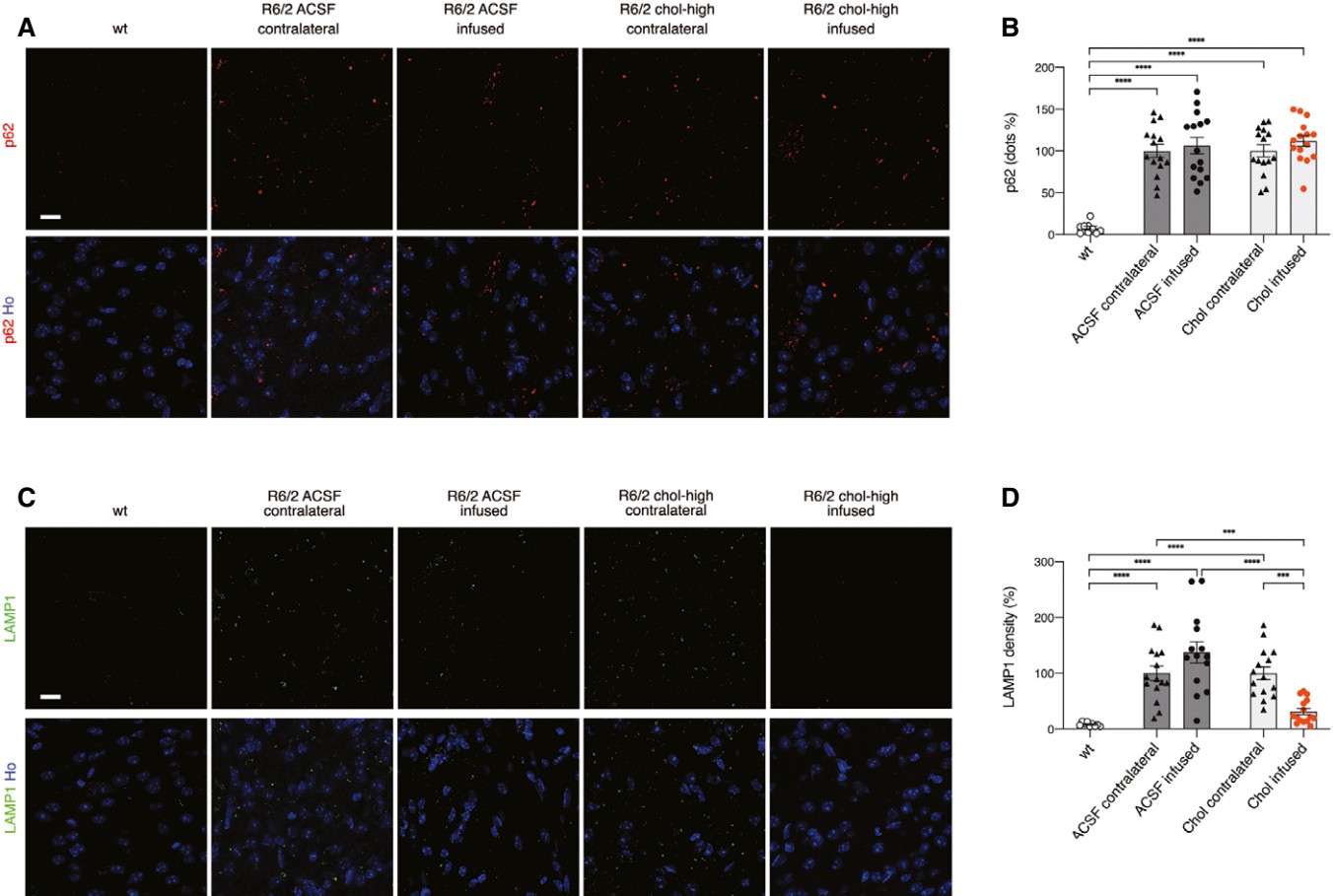

**Figure 6. Striatal infusion of the high dose of cholesterol promotes lysosomal clearance and autophagy in the striatum of HD mice.**

A–D   Representative confocal images showing p62 (A, red) or LAMP1 (C, green) in the infused and contralateral striatum on brain coronal sections from wt, R6/2 ACSF, and R6/2 chol-high mice (N = 3/group). Hoechst (Ho, blue) was used to counterstain nuclei. Quantification of dots for p62 (B) and density for LAMP1 (D) in the contralateral and infused striatum of R6/2 ACSF and R6/2 chol-high mice. 20 images from 3 sections in the middle of the striatum for each animal were acquired and analyzed. The data in (B and D) are shown as scatterplots with means ± standard error, and each dot corresponds to the value obtained from each image. Statistics: one-way ANOVA followed by Newman–Keuls multiple comparison tests (***P < 0.001; ****P < 0.0001). Scale bars in (A, C): 10 μm.

Source data are available online for this figure.

inhibitory transmission might bind cholesterol for their function, exerting a major effect on this circuit with respect to the excitatory one.

Exogenous cholesterol may also act by partnering specifically with cellular proteins and influencing cell physiology. A recent proteome-wide mapping of cholesterol-interacting proteins in mammalian cells detected more than 250 proteins that bind cholesterol. These proteins are involved in vesicular transport, degradation pathways, and membrane structure and dynamics, and many of them are linked to neurological disorders (Hulce et al, 2013). Among them, several membrane receptors bind cholesterol in cholesterol-enriched lipid rafts at the plasma membrane, and their interaction determines their function (Oddi et al, 2011; Guixà-González et al, 2017; Rahbek-Clemmensen et al, 2017; preprint: Casarotto et al, 2020). Once infused, exogenous cholesterol localizes at the plasma membrane and it might renormalize the stoichiometry between plasma proteins and receptors and rescue the impaired intracellular and receptor signaling in HD.

Cholesterol supplementation in animal models of Pelizaeus–Merzbacher disease and of multiple sclerosis results in a permissive environment for myelin repair, preventing disease progression (Saher et al, 2012; Berghoff et al, 2017). Changes in myelin also occur in different HD animal models (Xiang et al, 2011; Teo et al, 2016) and in patients showing pre-HD signs (Rosas et al, 2018). In those studies, overexpression of muHTT in primary oligodendrocytes was accompanied by reduced expression of cholesterol biosynthesis genes and myelin-binding protein in vitro (Xiang et al, 2011), and muHTT interfered with oligodendrocyte maturation in vivo (Rosas et al, 2018). However, striatal infusion of cholesterol failed to promote myelin repair in the R6/2 HD mouse model used here. Although restoration of myelin defects in this rapid, aggressive HD model was not observed under our experimental conditions, cholesterol administration may normalize myelin phenotypes in less aggressive HD murine models such as the YAC128 and knock-in mice (Huang et al, 2015; Teo et al, 2016).

In this study, we also found that striatal infusion of the high dose of cholesterol restored the primary defect of brain cholesterol biosynthesis in HD mice. Cholesterol biosynthesis, as judged by cholesterol precursor levels (Valenza *et al*, 2007b, 2010) or its synthesis rate (Shankaran *et al*, 2017), is significantly reduced in the striatum of HD mice before disease onset, as we confirm here. The significant increase in levels of cholesterol precursors, along with increased nuclear translocation of SREBP2 mainly in astrocytes, is consonant with an enhanced endogenous cholesterol biosynthesis in the striatum of R6/2 chol-high mice. This outcome is specific for the infused striatum and the highest dose of cholesterol, possibly indicating that excess exogenous cholesterol is converted into 24S-OHC in neurons that in turn stimulates endogenous synthesis in astrocytes (Janowski *et al*, 1999; Abildayeva *et al*, 2006). Accordingly, the level of 24S-OHC was also increased in the infused striatum of R6/2 chol-high mice in this work, supporting evidence that synthesis and catabolism are closely related in the disease state as well, as previously reported (Shankaran *et al*, 2017). This connection is in agreement with two recent studies showing that adeno-associated virus over-expressing Cyp46A1, the neuronal-specific enzyme for cholesterol conversion to 24S-OHC, increased lanosterol and desmosterol levels in the striatum of R6/2 mice and zQ175 mice (Boussicault *et al*, 2016; Kacher *et al*, 2019). However, we cannot exclude that different and independent mechanisms participate to increase cholesterol precursors and 24S-OHC levels in HD striatum following striatal infusion of cholesterol.

The increase in cholesterol precursors may also explain the reduction that we observed in muHTT aggregates in R6/2 mice. Lanosterol reverses protein aggregation in cataracts (Zhao *et al*, 2015), suppresses the aggregation and cytotoxicity of misfolded proteins linked to neurodegenerative diseases (Upadhyay *et al*, 2018), and promotes autophagy in Parkinson's disease models (Lim *et al*, 2012). In our study, clearance of muHTT in R6/2 chol-high mice may have been secondary to stimulation of mTORC1 activity (Narita *et al*, 2011; Lee *et al*, 2015). Several studies have established links between mTORC1 activation and cholesterol metabolism. For example, the expression of the active form of the mTORC1 regulator, Rheb, in the HD mouse brain ameliorates aberrant cholesterol homeostasis and increases autophagy (Lee *et al*, 2015). In addition, mTORC1 activation increases nuclear translocation of SREBP2 and sterol synthesis (Yecies & Manning, 2011; Owen *et al*, 2012). Moreover, an increase in lysosomal cholesterol has been reported to activate mTORC1 (Castellano *et al*, 2017), and cholesterol is reduced in HD lysosomes (Koga *et al*, 2011). How exactly the increased nuclear translocation of SREBP2 and decreased muHTT aggregates observed here are linked to mTORC1 activity and lysosomes is currently unknown. Exogenous cholesterol, which partially co-localizes with late endosomes, may also act on the axonal transport, which is compromised in HD (Gunawardena *et al*, 2003; Gauthier *et al*, 2004; White *et al*, 2015), and contribute to diminish organelles accumulation (Ferguson, 2018).

The fact that we found no significant reduction in aggregated muHTT with the TR-FRET assay (4C9-4C9 combination) suggests that cholesterol is not sufficient to degrade muHTT oligomers at the beginning of aggregation process. However, the significant decrease in muHTT species (assessed by TR-FRET with B27-MW1 combination) coupled with the reduction in macro-aggregates (assessed by immunofluorescence) and the absence of amyloid-like fibers (by EM) supports the hypothesis that cholesterol mitigates different steps of muHTT aggregation *in vivo* and that its targeted administration to the brain might be useful for reducing muHTT toxicity in HD.

In conclusion, we demonstrate a dose-dependent, disease-modifying effect of striatal infusion of cholesterol in HD mice. This work and our previous findings (Valenza *et al*, 2015b) support the hypothesis that reduced cholesterol biosynthesis contributes to disease pathogenesis and that cholesterol delivery to the HD brain is beneficial. Further studies will explore the potential for long-term cholesterol release in HD animal models with a longer lifespan and slower disease progression, enabling chronic treatment in older, symptomatic mice. In addition, with the aim of translating the delivery of cholesterol to the clinic, new brain-permeable nanoparticles have been developed (Belletti *et al*, 2018) that enable the controlled release of a higher cholesterol content to the brain. This advance may facilitate progress toward the goal of achieving the therapeutic dose identified here by systemic injection.

## Materials and Methods

### Colony management

All the *in vivo* experiments were carried out in accordance with Italian Governing Law (D.lgs 26/2014; Authorization n.324/2015-PR issued May 6, 2015 by Ministry of Health); the NIH Guide for the Care and Use of Laboratory Animals (2011 edition) and EU directives and guidelines (EEC Council Directive 2010/63/UE).

Our R6/2 colony lifespan was approximately of 13 weeks, and it was maintained through the male line exclusively (Mangiarini *et al*, 1996). Mice were weaned and then genotyped at 3 weeks of age ($\pm$ 3 days). Transgenic R6/2 males were paired with non-carrier females (B6CBAF1/J, purchased from Charles River). CAG repeat length and changes that could affect strain productivity, general behavior, litter size, pup survival, genotype frequency, phenotype were constantly monitored with a range between 200 and 250 CAGs. Mice were housed under standard conditions ($22 \pm 1$°C, 60% relative humidity, 12 h light/dark schedule, 3–4 mice/cage, with free access to food and water). After PCR genotyping (Mangiarini *et al*, 1996), male and female mice were included and randomly divided into experimental groups. Littermates were included as controls.

### Surgical implantation of osmotic mini-pumps

Avertin 100% was prepared diluting 5 g of 2,2,2-Tribromoethanol (Sigma-Aldrich, #T48402-25G) in 5 ml of 2-methyl-2-butanol (Sigma-Aldrich, #240486). Mice were deeply anesthetized using 15 µl of Avertin 2.5% per gram of body weight. Once responses to tail/toe pinches and intactness of the ocular reflex were assessed, scalp was shaved and mice were placed into a stereotaxic apparatus (2-Biological Instrument). A subcutaneous pocket was made on the back of the animals, in the midscapular area, to insert the osmotic mini-pump (Alzet, pump model 1004, #0009922). The brain infusion microcannula (brain infusion kit n°3, Alzet, #0008851), connected to the mini-pump through a catheter, was stereotaxically implanted into mice right striatum (stereotaxic coordinates 1.75 mm mediolateral, 0.5 mm anteroposterior, 3 mm dorsoventral; from Paxinos G and Watson C. The Rat Brain in Stereotaxic Coordinates. Academic Press, San Diego).

Following surgery, mice were removed from the stereotaxic apparatus and placed on a warm cover to awaken from anesthesia. The mini-pump infused at constant rate (0.11 µl/h) for 28 days a solution of artificial cerebrospinal fluid (ACSF); or methyl-β-cyclo-dextrin (Sigma-Aldrich, #M7439-1G) diluted in ACSF; or water-soluble methyl-β-cyclodextrin (MβCD)-balanced cholesterol (Sigma-Aldrich, #C4951-30MG) supplemented with 5 µM free cholesterol, diluted in ACSF. ACSF was prepared mixing two solutions (A and B) in a 1:1 ratio. Solution A was prepared by diluting 8.66 g of NaCl (Sigma-Aldrich, #S3014), 0.224 g of KCl (Sigma-Aldrich, #P9333), 0.206 g of $CaCl_2$ $2H_2O$ (Sigma-Aldrich, #C3881) and 0.163 g of $MgCl_2$ $6H_2O$ (Sigma-Aldrich, #M9272) in 500 ml of sigma water. Solution B was prepared by diluting 0.214 g of $Na_2HPO_4$ $7H_2O$ (Sigma-Aldrich, #S9390) and 0.027 g of $NaH_2PO_4$ $H_2O$ (Sigma-Aldrich, #S9638) in 500 ml of sigma water.

Assessment of post-operative pain and distress was performed using a specific table for pain scoring based on behavioral indicators of well-being and monitoring mice body weight (Lloyd & Wolfen-sohn, 1998).

## Behavioral tests

Mice behavior was evaluated from pre-symptomatic stages (5–6 weeks of age) until late-symptomatic stages of the disease (10–11 weeks of age). Animals were assigned randomly, and sex was balanced in the various experimental groups. All the behavioral analyses were performed in blind.

### Rotarod
Motor coordination and balance were evaluated on the rotarod test. Mice were first trained to walk on a rotating bar at constant speed of 4 rpm (apparatus model 47600, Ugo Basile), for 300 s. 1 h after this training phase, mice motor performance was evaluated in an accelerating task (from 4 to 40 rpm) over a 300-s period. For three consecutive days, mice performed three daily trials, with an inter-trial interval of 30 min. Latency to fall was recorded for each trial and averaged.

### Activity cage
Spontaneous locomotor activity was evaluated by the activity cage test, in presence of a low-intensity white light source. The animal was placed in the center of the testing, transparent, arena (25 cm × 25 cm) (2Biological Instrument) and allowed to freely move for an hour. Following 15 minutes of habituation, both hori-zontal and vertical motor activities were assessed by an automated tracking system (Actitrack software, 2Biological Instrument) connected to infrared sensors surrounding the arena. Total distance travelled, mean velocity speed, stereotyped movements, and numbers of rearings were evaluated. The % of time that mice explored the periphery or the center area of the was evaluated as a measure of anxiety-like behavior.

### Novel object recognition (NOR) test
Long-term memory was evaluated by the NOR test, using a gray-colored, non-reflective arena (44 × 44 × 44 cm). All phases of the test were conducted with a low-intensity white light source. In a first habituation phase, mice were placed into the empty arena for 10 min. The habituation phase was followed by the familiarization

one, in which two identical objects (A′ and A″) were presented to each animal for 10 min. Twenty-four hours later, during the testing phase, the same animals were exposed to one familiar object (A″) and a new object (B) for 10 min. A measure of the spontaneous recognition memory was represented by the index of discrimination, calculated as (time exploring the novel object − time exploring the familiar object) / (time exploring both objects) × 100. Mice explor-ing < 7 s were excluded from the analysis due to their inability to perform the task.

## PCA analysis

Principal component analysis (PCA) was performed using the R package ade4 (Pavoine et al, 2004).

## Gas chromatography–mass spectrometry (GC-MS) analysis for neutral sterols and 24S-hydroxycholesterol

To a screw-capped vial sealed with a Teflon-lined septum were added 50 µl of homogenates together with 1,000 ng of D4-lathosterol (CDN Isotopes, Canada), 100 ng of D6-lanosterol (Avanti Polar Lipids, USA), 400 ng of D7-24S-hydroxycholesterol (Avanti Polar Lipids, USA), and 50 µg of D6-cholesterol (CDN Isotopes, Canada) as internal standards, 50 µl of butylated hydroxytoluene (BHT) (5 g/l), and 25 µl of EDTA (10 g/l). Argon was flushed through to remove air. Alkaline hydrolysis was allowed to proceed at room temperature (22°C) for 1 h in the presence of 1 M ethanolic potassium hydroxide solution under magnetic stirring. After hydrolysis, the neutral sterols (cholesterol, lathosterol, and lanosterol) and oxysterols (24S-OHC) were extracted three times with 5 ml of hexane. The organic solvents were evapo-rated under a gentle stream of argon and converted into trimethylsilyl ethers with BSTFA-1% TMCS (Cerilliant, USA) at 70°C for 60 min. Analysis was performed by gas chromatography–mass spectrometry (GC-MS) on a Clarus 600 gas chromatograph (Perkin Elmer, USA) equipped with Elite-5MS capillary column (30 m, 0.32 mm, 0.25 µm. Perkin Elmer, USA) connected to Clarus 600C mass spectrometer (Perkin Elmer, USA). The oven temperature program was as follows: initial temperature 180°C was held for 1 min, followed by a linear ramp of 20°C/min to 270°C, and then a linear ramp of 5°C/min to 290°C, which was held for 10 min. Helium was used as carrier gas at a flow rate of 1 ml/min and 1 µl of sample was injected in splitless mode. Mass spectrometric data were acquired in selected ion monitor-ing mode. Peak integration was performed manually, and sterols were quantified against internal standards, using standard curves for the listed sterols (Leoni et al, 2017).

## Electrophysiological analysis

Experiments were performed on submerged brain slices obtained from adult mice (12 weeks of age) after 4-week long infusion of ACSF or cholesterol directly into the striatum. Animals were anes-thetized by inhalation of isoflurane and decapitated. The head was rapidly submerged in ice-cold (~4°C) and oxygenated (95% $O_2$ – 5% $CO_2$) cutting solution containing: Sucrose 70 mM, NaCl 80 mM, KCl 2.5 mM, $NaHCO_3$ 26 mM, Glucose 15 mM, $MgCl_2$ 7 mM, $CaCl_2$ 1 mM and $NaH_2PO_4$ 1.25 mM. Striatal coronal slices (300-µm-thick) were cut using a vibratome (DTK-1000, Dosaka EM, Kyoto, Japan) and allowed to equilibrate for at least 1 hour in a chamber filled

with oxygenated ACSF containing: NaCl 125 mM, KCl 2.5 mM, NaHCO$_3$ 26 mM, Glucose 15 mM, MgCl$_2$ 1.3 mM, CaCl$_2$ 2.3 mM and NaH$_2$PO$_4$ 1.25 mM. The slices collected from the hemisphere ipsilateral to the infusion site were transferred to a submerged-style recording chamber at room temperature (~ 23–25°C) and were continuously perfused at 1.4 ml/min with ACSF. The chamber was mounted on an E600FN microscope (Nikon) equipped with 4× and 40× water immersion objectives (Nikon) and connected to a near-infrared CCD camera for cells visualization.

Data were obtained from striatal projection medium spiny neurons (MSNs) using the whole-cell patch-clamp technique in both voltage- and current-clamp mode. The patch pipette was produced from borosilicate glass capillary tubes (Hilgenberg GmbH) using a horizontal puller (P-97, Sutter instruments) and filled with an intracellular solution containing: Cs-methanesulphonate 120 mM, KCl 5 mM, CaCl$_2$ 1 mM, MgCl$_2$ 2 mM, EGTA 10 mM, Na$_2$ATP 4 mM, Na$_3$GTP 0.3 mM, Hepes 8 mM and lidocaine N-ethyl bromide 5 mM (added to inhibit firing by blocking intracellularly the voltage-sensitive Na$^+$ channels) (pH adjusted to 7.3 with KOH). Spontaneous excitatory postsynaptic currents (sEPSCs), mediated by the activation of ionotropic glutamate receptors, were recorded from MSNs at a holding potential of $-70$ mV, whereas spontaneous inhibitory postsynaptic currents (sIPSCs), elicited by the activation of GABA$_A$ receptors, were assessed at a holding potential of 0 mV. The signals were amplified with a MultiClamp 700B amplifier (Molecular Devices) and digitized with a Digidata 1322 computer interface (Digitata, Axon Instruments Molecular Devices, Sunnyvale, CA). Data were acquired using the software Clampex 9.2 (Molecular Devices, Palo Alto, CA, U.S.A.), sampled at 20 kHz and filtered at 2 kHz.

The off-line detection of spontaneous postsynaptic currents (sPSCs) were performed manually using a custom-made software in Labview (National Instruments, Austin, TX, U.S.A.). The amplitudes of sPSCs obeyed a lognormal distribution. Accordingly, the mean amplitude was computed as the peak of the lognormal function used to fit the distribution. Intervals (measured as time between two consecutive sPSCs) for spontaneous events were distributed exponentially and the mean interval was computed as the tau ($\tau_{interval}$) value of the mono-exponential function that best fitted this distribution. The reciprocal of $\tau$ ($1/\tau$) is the mean of the instantaneous frequencies of sPSCs. Furthermore, the analysis of the membrane capacitance ($C_m$) and the input resistance ($R_{in}$) was performed using Clampfit 10.2 (Molecular Devices, Palo Alto, CA, U.S.A.). $C_m$ was estimated from the capacitive current evoked by a -10 mV pulse, whereas $R_{in}$ was calculated from the linear portion of the I-V relationship obtained by measuring steady-state voltage responses to hyperpolarizing and depolarizing current steps.

## Immunohistochemistry analysis

Mice were anesthetized by intraperitoneal injection of Avertin 2.5 % and transcardially perfused with PFA 4 %. Brains were post-fixed overnight in the same solution at 4°C and then in 30% sucrose to prevent ice crystal damage during freezing in OCT.

Immunohistochemistry was performed on 15 μm coronal sections. Epitopes were demasked at 98°C with NaCitrate 10 mM and then slices were incubated with the following primary antibodies for 3 h at RT: rabbit anti-SREBP2 (1:2,000; gift by T. Osborne) (Seo *et al*, 2012),

mouse anti-SREBP2 (1:100; Ls-Bio, LS-C179708), rabbit anti-DARPP32 (1:100; Cell Signalling, 2306), mouse anti-NeuN (1:100; Millipore, MAB377), rabbit anti-NeuN (1:500; Abcam, AB104225), rabbit anti-GFAP (1:250; Dako, Z0334), mouse anti-Huntingtin clone EM48 (1:100; Millipore, MAB5374), rabbit anti-p62 (1:100; Abcam, AB109012) or rat anti-LAMP1 (1:50; Santa Cruz, SC19992), rabbit anti-TGN46 (1:60; Abcam, ab16059), rabbit anti-Rab9A (1:50; Euro-Clone, BK5118S-CST), rabbit anti-calnexin (1:100; Life technologies, PA534754), mouse anti PMCA-ATPase (1:500; Thermo Fisher Scientific, MA3-914). Anti-rabbit or anti-mouse Alexa Fluor 568-conjugated goat secondary antibodies (1:500; Invitrogen), anti-rabbit Alexa Fluor 633-conjugated goat secondary antibodies (1:500; Invitrogen) or anti-mouse Alexa Fluor 488-conjugated goat secondary antibodies (1:500; Invitrogen) were used for detection (1 h at RT) depending on the primary antibodies. Sections were counterstained with the nuclear dye Hoechst 33258 (1:10,000, Invitrogen) and then mounted under cover slips using Vectashield (Vector Laboratories).

## Image acquisition and quantification

Confocal images were acquired with a LEICA SP5 laser scanning confocal microscope. Laser intensity and detector gain were maintained constant for all images and 3–10-z steps images were acquired.

To count aggregates in the different brain areas 34 images/mice taken from three R6/2-ACSF and three R6/2-chol mice were made at 40×. For the striatum 18 images/animal were analyzed from 9 sections throughout the entire striatum. For the cortex, 10 images for each animal were analyzed from 3 sections and for the hippocampus, 6 images for each animal/condition were analyzed from 3 sections. To quantify the number of aggregates, ImageJ software was used to measure the fluorescence. Images were divided into three-color channels and the same global threshold was set. In both R6/2-ACSF and R6/2-chol mice, the total number of aggregates in the infused hemisphere was normalized to the total number of aggregates in the contralateral hemisphere. To count the number of aggregates in the nucleus of DARPP32 or GFAP-positive cells, the NIS software was used.

To quantify the amount of SREBP2 inside the nucleus, images were acquired at 40× and were segmented using the NIS software. A threshold was applied to both channels and the intensity ratio of SREBP2/Hoechst was measured.

To count the dots of p62 and LAMP1 in the different brain areas 20 images/mice taken from three R6/2-ACSF and three R6/2-chol mice were made at 40×. For the striatum 10 images/animal were analyzed from 3 sections in the middle of the striatum. For the cortex, 10 images for each animal were analyzed from 3 sections. To quantify the number of dots, ImageJ software was used to measure the fluorescence. Images were divided into three-color channels and the same global threshold was set. In both R6/2-ACSF and R6/2-chol mice, the total number of dots in the infused hemisphere was normalized to the total number of dots in the contralateral hemisphere.

## Electron microscopy

### Sample preparation

Mice were anesthetized by intraperitoneal injection of 10 mg/ml Avertin (Sigma) and transcardially perfused using a fixative solution

of: 2.5% glutaraldehyde (#16220 Electron Microscopy Sciences (EMS), Hartfield, PA), and 2% paraformaldehyde (P16148 EMS) in sodium cacodylate buffer 0.15 M (pH 7.4) (#12300 EMS). Brains were removed and post-fixed for additional 24 h at 4°C. Brains were cut in 100 μm-thick coronal slices by using a Leica VT1000S vibratome. Sections were collected in sodium cacodylate buffer 0.1 M and striatum, and corpus callosum were manually dissected for staining and embedding. Samples were then washed with cold sodium cacodylate buffer 0.1 M and then post-fixed in a reduced osmium solution (i.e., 1.5% potassium ferrocyanide (#20150 EMS) with 2% osmium tetroxide (#19170 EMS) in 0.15 M cacodylate buffer, for 1 h in ice. After the first heavy metal incubation, the tissues were washed with ddH$_2$O at room temperature and then placed in the 0.22 μm-Millipore-filtered 1% thiocarbohydrazide (TCH) (#21900 EMS) in ddH$_2$O solution for 20 min, at room temperature. Tissues were then rinsed again in ddH$_2$O and incubated in 2% osmium tetroxide in ddH$_2$O for 30 min, at room temperature. After several washings at room temperature in ddH$_2$O, they were then placed in 1% uranyl acetate (aqueous), overnight at 4°C. Samples were washed and then incubated *en-bloc* in Walton's lead aspartate solution (0.066 gr lead nitrate (#17900 EMS) dissolved in 10 ml of 0.003 M aspartic acid solution, pH 5.5) at 60°C for 30 min. The tissues were still washed and then dehydrated with an ethanol series and finally placed in anhydrous ice-cold acetone for 10 min. Infiltration was performed with acetone (#179124 Sigma-Aldrich)—Durcupan ACM® (#14040 EMS) mixture with 3:1 volume ratio for 2 h, then 1:1 overnight. The tissues were left for 2 h in pure resin and then embedded in Durcupan ACM® resin and placed in a 60°C oven for 48 h for polymerization.

### TEM imaging

Ultrathin sections 70 nm-thick were prepared by an UltraCut E ultramicrotome (Reichert) and collected on TEM copper grids, which were then observed by a LEO 912AB microscope (Carl Zeiss), equipped with a thermionic W electron source and operating at an acceleration voltage of 100 kV. For quantitative analyses, images with a resolution of 1,024 × 1,024 pixels were acquired using a bottom mount Esivision CCD-BM/1K system (ProScan Camera). Quantitative measurements were performed by ImageJ 1.47v, and measuring the following parameters: total SVs density as the number of SVs divided by the pre-synaptic area (μm²), number of docked vesicles per active zone (AZ) length (μm), pre-synaptic area (μm²), active zone (AZ) length (μm) and PSD area (nm²) and PSD length (nm). For myelin analyses, we measured the G-ratio, as the diameter of the axon/outer diameter of the myelinated fiber (of at least 300 myelinated axons in 3 mice group) and the myelin periodicity that was measured as the mean distance between two major dense lines, in at least 45 randomly chosen myelin sheaths in 3 mice group.

### FIB-SEM imaging and ion cutting

The following procedure was used to mount specimens with the aim to minimize theirs electrical charging during the FIB-SEM imaging and ion cutting. Resin blocs were mounted on aluminum specimen pins and trimmed with a glass knife using an ultramicrotome, to expose the tissue on all four sides. Silver paint (#16031 Ted Pella, Redding, US) was used to electrically ground the edges of the tissue block to the aluminum pin. The entire specimen was then coated

with a thin layer of gold by means of a Cressington 208-HR sputter coater (Cressington Scientific Instruments, Watford, UK) equipped with a pure gold target (Ted Pella, Redding, US), to finely mount it into the SEM chamber in view of the FIB-SEM imaging. The sample 3D ultrastructural imaging was performed by using a Thermo Scientific Helios G4 Dual Beam (Eindhoven, NL) being this instrument constituted by the combination of a high-resolution SEM equipped with a Schottky field emission gun and a focused gallium ion beam. First, the region of interest was chosen on the surface of the tissue block, and then, a protective layer of platinum was deposited on top of the area to be imaged using a gallium ion beam with 30 kV of acceleration voltage. Initially, a rough cross-section was milled by a 9.1 nA ion beam current, and used as window for SEM imaging. The exposed surface of this cross-section was finely polished by progressively lowering the ion beam current down to 0.44 nA and keeping the acceleration voltage at 30 kV. Afterward, layers from the finely polished cross-section were successively milled by the gallium ion beam, again using a current of 0.44 nA and an acceleration voltage of 30 kV. To remove each layer, the ion beam was continuously moved closer to the surface of the cross-section by increments of 25 nm. After each slice ion cutting, the milling process was automatically paused, and the newly exposed surface was imaged with a 2 kV acceleration voltage and 0.2 nA electron beam current using the through-the-lens backscattered electron detector (TLD-BSE). The slicing and imaging processes were continuously repeated, and a long series of images were acquired in a automated procedure. SEM images of 2048 × 1768 pixels were acquired with voxel size of (3 × 3 × 25) and (4 × 4 × 25) nm, depending on the SEM magnification chosen.

### 3D reconstruction, rendering, and analysis

Serial SEM images were assembled into volume files aligned using the FiJi software (Schindelin *et al*, 2009) plugin called linear stack alignment with SIFT (Lowe, 2004). Following the images acquisition, recording, and alignment, the 3D shape of samples peculiar features (in our case excitatory and inhibitory synapses) was reconstructed layer by layer by careful segmentation. For performing the latter, and the 3D model generation, electron microscopy image stacks were then converted to 8-bit grayscale tiff format images and manually segmented using AMIRA software package (Thermo Scientific, Eindhoven, NL). Three-dimensional structures in image stacks containing hundreds or thousands of 2D orthoslices were traced individually in each plane and automatically surface rendered. The excitatory and inhibitory synapse density (n° of synapse/μm³) was finally measured by using Ilastik-0.5.12 software.

### Sample preparation for pre-embedding immunogold labeling

Mice were anesthetized by intraperitoneal injection of 10 mg/ml Avertin (Sigma) and transcardially perfused using pH-shift formaldehyde (Berod *et al*, 1981): 4% paraformaldehyde (P16148 EMS) 0.1 M sodium acetate buffer, pH 6.0, followed by the same fixatives in 0.1 M sodium carbonate buffer, pH 10.5. Brains were removed and post-fixed for additional 24 h at 4°C and 100 μm-thick coronal slices were cut by using a Leica VT1000S vibratome. Sections were collected in 0.1 M sodium carbonate buffer, pH 10.5, and striatum was manually dissected. Striatal sections were incubated with mouse monoclonal antibodies (EM48) 1:50 (MAB5374-Millipore) that reacts with human huntingtin protein (both native

and recombinant protein) in PBS containing 1% NGS 48 hours at 4°C. After rinsing in PBS, samples were incubated with goat anti-mouse secondary antibodies (1:50) conjugated to 10 nm gold particles (Jackson ImmunoResearch) in PBS with 2% NGS overnight at 4°C. After rinsing in PBS, sections were osmicated in 1% $OsO_4$ in ddH$_2$O and stained overnight in 2% aqueous uranyl acetate. All sections used for electron microscopy (EM) were dehydrated in ascending concentrations of ethanol and acetone/eponate 12 (1:1) and embedded in Eponate 12 (#14120 EMS). Ultrathin sections (70 nm) were cut using an UltraCut E ultramicrotome (Reichert) and placed on TEM copper grids. Thin sections were counterstained with 1% aqueous uranyl acetate for 5 min followed by 1% lead citrate in ddH$_2$O for 2 min and examined using a LEO 912AB microscope (Carl Zeiss), equipped with a thermionic W electron source and operating at an acceleration voltage of 100 kV. Images were acquired at a resolution of 1,024 × 1,024 pixels using a bottom mount Esivision CCD-BM/1K system (ProScan Camera).

### Sample preparation for post-embedding immunogold labeling

Mice were anesthetized by intraperitoneal injection of 10 mg/ml Avertin (Sigma) and transcardially perfused using pH-shift formaldehyde (Berod *et al*, 1981): 4% paraformaldehyde (P16148 EMS) 0.1 M sodium acetate buffer, pH 6.0, followed by the same fixatives in 0.1 M sodium carbonate buffer, pH 10.5. Brains were removed and post-fixed for additional 24 h at 4°C and 100 μm-thick coronal slices were cut by using a Leica VT1000S vibratome. Sections were collected in 0.1 M sodium carbonate buffer, pH 10.5, and striatum was manually dissected and post-fixed with 1% $OsO_4$ in ddH$_2$O and stained with 0.5% uranyl acetate. Samples were dehydrated in ascending concentrations of ethanol and acetone/eponate 12 (1:1) and embedded in Eponate 12 (#14120 EMS). Striatal sections were cut in a UltraCut E ultramicrotome (Reichert). Formvar carbon-coated nickel grids with 70 nm ultrathin sections were processed for GABA immunolabeling. After 5 min incubation in TBST pH 7.6, grids were incubated with rabbit antiserum against GABA (Sigma A2052, 1:10.000 in TBST) overnight at RT in a moist chamber. After the incubation, grids were washed 3 × 10 min TBST pH 7.6, followed by TBST pH 8.2 for 5 min. Grids were incubated for 2 h in goat anti-rabbit IgG conjugated to 12 nm colloidal gold (Jackson ImmunoResearch) diluted 1:20 in TBST pH 8.2. They were then washed twice in TBST pH 7.6 and rinsed in deionized water. After that, the grids were contrast-stained with 1% aqueous uranyl acetate for 5 min followed by 1% lead citrate in ddH$_2$O for 2 min. Sections were examined using a LEO 912AB microscope (Carl Zeiss), equipped with a thermionic W electron source and operating at an acceleration voltage of 100 kV. Images were acquired at a resolution of 1,024 × 1,024 pixels using a bottom mount Esivision CCD-BM/1K system (ProScan Camera).

### FRET analysis

TR-FRET assays were performed as described previously (Weiss *et al*, 2009). Briefly, 15 μl of each homogenate was transferred to a low volume 384-well plate (Greiner) in serial dilutions starting from a defined concentration (4 μg/μl), 3 μl of antibody cocktail was then added. MuHTT aggregates were measured with 4C9-Tb/4C9-Alexa 647, using 1.93 ng/μl of 4C9-Tb and 2 ng/μl of 4C9-Alexa 647-labeled antibodies. Soluble muHTT and other muHTT

### The paper explained

#### Problem

Cholesterol is fundamental for several activities of the brain. Peripheral cholesterol is not able to reach this organ due to the presence of the blood–brain barrier; thus, the majority of cholesterol found in the brain is synthetized locally. *De novo* synthesis of cholesterol is reduced in the Huntington's disease (HD) brain before the clinical disease onset and strategies aimed at providing cholesterol to the HD brain may be beneficial. However, the identification of the therapeutic dose of cholesterol that must reach the brain to have a maximum benefit on the multiple disease-related phenotypes is still unknown and needs to be qualified for a translational perspective.

#### Results

Here we infused three escalating doses of cholesterol in the brain of HD mice by the use of osmotic mini-pumps, and we identified the dose that is able to reverse both cognitive and motor abnormalities. We found that cognitive decline was prevented by all the three tested doses, while motor dysfunction was reversed only with the highest dose. Moreover, exogenous cholesterol acted at multiple levels by normalizing a plethora of disease-related dysfunctions including those linked to synapse function and morphology and aggregation of mutated Huntingtin.

#### Impact

Our work highlights the therapeutic dose of exogenous cholesterol capable of improving behavioral, synaptic, and neuropathological abnormalities in HD. This knowledge creates a solid foundation for developing new therapeutic strategies based on cholesterol to fight this disease.

species level was measured with 2B7-Tb/MW1-D2, using 1 ng/μl of 2B7-Tb and 10 ng/μl of MW1-D2-labeled antibody. TR-FRET measurements were routinely performed following overnight incubation at 4°C using an EnVision Reader (Perkin Elmer). Values were collected as the background subtracted ratio between fluorescence emission at 665 nm and 615 nm where the background signal corresponds to the ratio (665/615) measured for the antibodies in lysis buffer. The points in the graphs correspond to the averages of the background subtracted fluorescence ratio relative to the sample. The dilution points of each sample were fitted in a 4 parameters function that describes the curves. The obtained values were also expressed as (1/IC50) per μg of total protein for both assays.

### Statistics

Prism 6 (GraphPad software) was used to perform all statistical analyses. Data are presented as means ± standard error of the mean (s.e.m.). Grubbs' test was applied to identify outliers. For each set of data to be compared, we determined whether data were normally distributed or not to select parametric or not parametric statistical tests. The specific statistical test used is indicated in the legend of all results figures. Differences were considered statistically if the $P$-value was < 0.05. G-power software was used to predetermine group allocation, data collection, and all related analyses. For animal studies, mice were assigned randomly, and sex was balanced in the various experimental groups; animals from the same litter were divided in different experimental groups; blinding of the investigator was applied to *in vivo* procedures and

all data collection. Table EV3 summarizes all the trials and read-outs performed.

## Data and software availability

This study does not include data deposited in public repositories.

**Expanded View** for this article is available online.

## Acknowledgments

The authors acknowledge the scientific and technical assistance of Dr. Chiara Cordiglieri, responsible of the INGM Imaging Facility (Istituto Nazionale Genetica Molecolare—INGM, Milan, Italy); Dr. Alex Costa, Dr. Nadia Santo and the NOLI-MITS advanced imaging facility established by the University of Milan. The authors also acknowledge Prof. Timothy F. Osborne (Department of Medicine, Johns Hopkins University, Baltimore, Maryland, USA), who supplied the antibody anti-SREBP2. This work was supported by Telethon Foundation, Italy (# GGP17102), the EU projects Neuromics (FP7 #305121), and JPND Research CircProt (643417) to E.C.; by KAUST Baseline funding to A.F.; by the Italian Ministry of Education, University and Research (MIUR) Dipartimenti di Eccellenza Program (2018–2022)—Dept of Biology and Biotechnology "L. Spallanzani", University of Pavia to G. Biella, F. Talpo and C.M. F. Talpo was supported by Fondazione Umberto Veronesi. The visual abstract was created with BioRender.com.

## Author contributions

EDP, MVal, and EC conceived the study; EDP and GBir performed *in vivo* experiments, including surgical implantation of osmotic mini-pumps and behavioral analysis; GBir and MVal performed immunostaining experiments and provided confocal images and quantification; EV prepared samples for the TEM and FIB-SEM imaging and performed the TEM imaging; AF and ES performed the FIB-SEM imaging; EV and AF analyzed the TEM and FIB-SEM data; CM, FTal, and GBie performed and analyzed the electrophysiological recordings; CCac, FTar, and VL performed and analyzed mass spectrometry experiments; LP, CCar, MVer, and AC performed and analyzed TR-FRET experiments; VDB performed the PCA analysis; PC provided reagents/tools and suggestions for experiments regarding muHTT clearance and autophagy; MVal and GBir collected study data and performed statistical analyses; MVal and EC oversaw and coordinated responsibility for all research activities and their performance and provided experimental advice throughout the work. EC secured the funding, the collaborations, and the execution of the entire project. MVal, GBir, and EC wrote the paper that has been edited and reviewed by all authors.

## Conflict of interest

The authors declare that they have no conflict of interest.

## For more information

(i)  Website of the laboratory: http://www.cattaneolab.it/?lang=en
(ii) Website of the European Huntington Disease Network, a nonprofit research network committed to advancing research, facilitating the conduct of clinical trials, and improving clinical care in Huntington's disease: http://www.ehdn.org/

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
