## [Review Process File · EMBO Molecular Medicine]

Striatal infusion of cholesterol promotes dose-dependent behavioral benefits and exerts disease-modifying effects in Huntington's disease mice

Giulia Birolini, Marta Valenza, Eleonora Di Paolo, Elena Vezzoli, Francesca Talpo, Claudia Maniezzi, Claudio Caccia, Valerio Leoni, Franco Taroni, Vittoria Bocchi, Paola Conforti, Elisa Sogne, Lara Petricca, Cristina Cariulo, Margherita Verani, Andrea Caricasole, Andrea Falqui, Gerardo Biella, and Elena Cattaneo

DOI: [10.15252/emmm.202012519](https://doi.org/10.15252/emmm.202012519)

Corresponding authors: [Marta Valenza \(marta.valenza@unimi.it\)](mailto:marta.valenza@unimi.it), [Elena Cattaneo \(elena.cattaneo@unimi.it\)](mailto:elena.cattaneo@unimi.it),

Review Timeline:

Submission Date:	15th Apr 20
Editorial Decision:	6th May 20
Revision Received:	21st Jul 20
Editorial Decision:	7th Aug 20
Revision Received:	13th Aug 20
Accepted:	26th Aug 20

Editor: *Zeljko Durdevic*

Transaction Report:

6th May 2020

Dear Prof. Cattaneo,

Thank you for the submission of your manuscript to EMBO Molecular Medicine. We have now heard back from the three referees who agreed to evaluate your manuscript. As you will see from the reports below, the referees acknowledge the interest of the study. However, they raise some concerns that should be addressed in a major revision of the present manuscript. Addressing the reviewers' concerns in full will be necessary for further considering the manuscript in our journal.

Acceptance of the manuscript will entail a second round of review. Please note that EMBO Molecular Medicine encourages a single round of revision only and therefore, acceptance or rejection of the manuscript will depend on the completeness of your responses included in the next, final version of the manuscript. For this reason, and to save you from any frustrations in the end, I would strongly advise against returning an incomplete revision.

We realize that the current situation is exceptional on the account of the COVID-19/SARS-CoV-2 pandemic. Therefore, please let us know if you need more than three months to revise the manuscript.

I look forward to receiving your revised manuscript.

***** Reviewer's comments *****

Referee #1 (Remarks for Author):

Previous work from this group demonstrated that the systemic administration of brain permeable, cholesterol-loaded, nanoparticles resulted in a positive outcome (rescue/prevention of phenotype) in a mouse model of HD. In this work, the authors directly injected cholesterol in the striatum. By taking this type of approach the authors wanted to: a) identify an appropriate treatment (dose) regime, and b) gain mechanistic insights

Regarding the first objective, the data presented are, at the dose and time of exposure investigated, clear and easy to interpret: the four week long treatment had a positive outcome in several of the HD-like deficits, motor, cognitive, at the electrophysiological level and in terms of certain cell biological variables (number of synaptic vesicles, gabaergic terminals). Despite these interesting results, there are a number of questions that I would need the authors to answer before I can support the publication of this work in EMBO Mol. Med.

1) Does this treatment remain effective when applied for longer periods? This is of relevance as a) patients may need chronic treatment, and b) chronic exposure to cholesterol may be toxic to cells, especially if it accumulates in lysosomes. This last point brings me to the next question:

2) Where does the exogenous cholesterol partition?. Authors should attempt to determine the membrane/intracellular localisation of the exogenous cholesterol (Bodipy-labelled cholesterol followed by LM?, biotinylated, non-cytolytic Perfringolysin O followed by EM?). It would be quite important to have more information on the localisation, especially to demonstrate that the exogenous cholesterol is present in the cellular domains where the endogenous cholesterol localises (plasma membrane, trans Golgi, late endosomes) and absent in ER and lysosomes, organelles where cholesterol accumulation is known to impair function.

3) Would the doses utilised in 7 week-old mice be equally effective in older mice? I am aware this question cannot be addressed experimentally with this strain of mice so I just ask authors to comment on this. This question is in my opinion pertinent for the simple reason that the average onset of this disease in humans is 40 years of age, which in the strain of mice utilised here would imply 10-14 month-old mice (not the 2 month-old mice used here).

4) How long do the beneficial effects remain after infusion is terminated? This is an important aspect to be considered for future plans of applicability of this approach: continuous or discontinuous infusion ? Did authors not allow some mice to live longer after the end of the 4 weeks of treatment?

I also have some questions concerning the second objective of this work, on the mechanism involved in the improved phenotype.

1) What is the contribution of increased endogenous cholesterol synthesis to the improved phenotype, compared to the effect of exogenous cholesterol? Co-infusion with an inhibitor of cholesterol synthesis (or a viral particle-mediated localised knockdown of SREBP2) could help to address this question. Although the (improved) phenotype due to cholesterol infusion is clear I consider important to address the contribution of the exogenous and endogenous cholesterol. The observation that exogenous cholesterol (Bodipy signal) colocalises with LAPM1 does not prove that exogenous cholesterol is responsible for the improved phenotype (see below).

2) How does increased biosynthesis occur in a high cholesterol content milieu? Authors assume that increased synthesis is due to high 24S hydroxycholesterol (from the exogenous cholesterol), in turn leading to SREPB2 activation in glial cells. An alternative view, even a simpler view, is that increased synthesis is the consequence of a further reduction of endogenous cholesterol due to cyclodextrin that becomes "free" in the injected area (once in the membrane cholesterol moves out of cyclodextrin ring), or direct activation of (endogenous) cholesterol catabolic activity (endogenous 24S OH Cholesterol) due to the presence of an ectopic (external) pool. Both

conditions can result in reduced endogenous cholesterol (the increase 24 S hydroxysterol from the endogenous pool) and consequently activation of SREBP2 pathway, which would be consistent with the canonical pathway of SREBP2 activation.

3) It would be quite unique if cholesterol accumulation in lysosomes improves its degradative function: the Niemann Pick type C is a clear example of neurological (in addition to systemic) disorder due to accumulation of cholesterol in lysosomes. Therefore, the accumulation of cholesterol in LAMP1 organelles reported here is more likely synonym of lysosomal dysfunction. On the other hand, the result of reduced aggregated Huntingtin is clear, implying that improved degradation could be due to better degradative activity by the proteasome and/or phagosome. Authors should do more to determine how exogenous cholesterol increases degradation (does the increase in lanosterol enhance mTORC1 activity?). Authors should be very cautious associating improved lysosomal function with cholesterol accumulation (quoting authors: Colocalization of LAMP1 and bodipy in the striatum of R6/2 mice (Fig 6E) suggests that cholesterol may act directly on lysosomes to enhance muHTT clearance)

Referee #2 (Remarks for Author):

This is an interesting and thorough manuscript done by an expert team. The experiments are well designed and the conclusions are meaningful. I have a number of concerns regarding interpretations and conclusions which should be addressed.

Figure 1D: I am concerned with the baseline Rotarod performance of the mice at 5 weeks (preimplantation). Although not statistically significantly different, the baseline performance of the mice destined to be in the high dose group is greater than the control R6/2 mice at a magnitude which approximately persisted throughout the trial. What were the starting weights of the R6/2 mice in the different groups at the outset and through the trial? Were they the same sex? Were there any differences in end-stage or mortality?

Figure 2A-B: From the methods, the cannula was inserted in the right hemisphere and the studies were done on the right side. Given the remarkable effects noted on figure 1, a question is why the magnitude of the behavioral data was as profound as it was while the infusion and biochemical effects were only unilateral? It appears from the data in supplementary figure 3 that there is no increase in cholesterol levels in the contralateral hemisphere. How do the authors explain the remarkable impact given that HD is a bilateral disease and the therapy and the biochemical changes are unilateral?

Figure 2C: Please clarify in the figure to which group is the middle bar different, WT and/or high Chol?

Figure 4f The morphology of stained cells appear potentially as neuronal. Please perform stain with neuronal and astrocytic marker in these slices. The cells demonstrating nuclear translocation look more like the cells in 4H than the ones in 4I. Please show what a WT control slice looks like. What would a WT slice look with cholesterol infusion (i.e. translocation)?

The data in figure 5 is confusing. EM48 staining is reduced in the infused region. It is my

understanding that the pattern of EM48 staining represents the mHTT aggregates. The data in figure 5N and O points to the soluble as the mHTT species that is impacted. Please explain and clarify in the text.

As noted in the discussion, demonstrating a link between the therapy and mTORC1 would add mechanistic insight to the study.

Referee #3 (Remarks for Author):

This study examined the effects of infusion of cholesterol into the striatum of HD R6/2 mice to determine if cholesterol deficits and phenotypes could be altered. The authors found increase in cholesterol synthesis, improvement in cognitive and motor behaviors and synaptic activity, and reduction in aggregates and mutant huntingtin levels. The work builds on earlier findings by some of the authors, is very carefully executed, and very nicely presented. These data in mice have considerable significance for considering cholesterol supplementation as a potential therapy for HD patients and will be of a great interest to a wide scientific community. The interpretation of some results is weak as pointed out below. The underlying mechanisms for how elevated cholesterol in astrocytes improves neuronal function in different subcellular compartments is loosely connected.

Fig 1. The behavioral benefits of cholesterol infusion were achieved using ipsilateral delivery to the striatum. This is somewhat surprising. The authors should comment on why unilateral infusion was sufficient and if bilateral infusion would have allowed for a lower dose to be used for many of the studies.

Fig. 1 Corpus callosum contains some fluorescent cholesterol and cortex was also infused. How do the benefits attributed to striatum relate to improved corticostriatal connectivity particularly excitatory activity? Were other brain areas also fluorescent?

Fig 2 G. The total number of excitatory synapses based on FIB-SEM was not changed by cholesterol infusion.

Fig 2 H-1. The analysis of synaptic vesicle depletion in HD and increase of synaptic vesicles after cholesterol infusion should consider other variables. Did the authors take into account the total number of synaptic vesicles, length of the synaptic contact and the number of vesicles at non synaptic membranes in the terminal as well?

If these are excitatory synapses, the synaptic contact should look asymmetric. The R6/2 example (upper right) does not look asymmetric in the EM image as it is depicted in the adjacent drawing. The scale bar for H and I seems off and should be checked. Synaptic vesicles are typically about 30-50 nm.

Results for Fig 2 G, Fig 3, Fig 4, and Fig 5 are not dose dependent and therefore somewhat misleading given the title of the paper.

Fig 4. It is not clear how increase in cholesterol in astrocytes affects neurons. The legend and labeling do not agree. DARPP32 is not labeled in the figure.

Fig 5 F Astrocyte size and branching looks increased in the image of the high cholesterol infused side. Are the cells reactive? Has cholesterol increased GFAP levels? Are microglia reactive?

Fig 5H legend describes EM 48 in the striatal neuropil but only cell bodies are shown.

In the R62 ACSF treatment in left column images -the fibrous structure described near the nuclear membrane is not clear but underneath that labeling fibrillar structure is more apparent.

Fig 5N. There is no change in aggregate load with TR FRET 4C9-4C9 aggregate assay with high dose cholesterol infusions compared to marked reduction in aggregates seen by immunostaining. What accounts for this apparent discrepancy? Small sample size or the assay itself?

Fig 6. The presence of perinuclear Lamp1 lysosomes is interpreted as clearance by lysosomes. It is not clear what they are clearing. The authors should consider the other possibility that the HD mutation impairs anterograde transport that results in retrograde accumulation of lysosomes. Depletion of synaptic vesicles may also be due to impaired anterograde transport.

High dose cholesterol did not rescue myelin depletion in the R62 mice. Were oligodendroglia markers checked?

Authors suggest from their data that there is increased availability of cholesterol in glial cells. How does this in turn enhance neuronal function? Could the cholesterol upoad by glia be a reactive/phagocytic response?

We are grateful to the reviewers for their comments/suggestions and for giving us the opportunity to re-submit a new version of the manuscript. Please find below our response to the 34 points raised by the three reviewers.

We hope that the revised version of this manuscript will now be judged suitable for publication to EMBO Molecular Medicine.

Best, Elena Cattaneo and Marta Valenza

REFEREE #1 (REMARKS FOR AUTHOR):

Previous work from this group demonstrated that the systemic administration of brain permeable, cholesterol-loaded, nanoparticles resulted in a positive outcome (rescue/prevention of phenotype) in a mouse model of HD. In this work, the authors directly injected cholesterol in the striatum. By taking this type of approach the authors wanted to: a) identify an appropriate treatment (dose) regime, and b) gain mechanistic insights

Regarding the first objective, the data presented are, at the dose and time of exposure investigated, clear and easy to interpret: the four week long treatment had a positive outcome in several of the HD-like deficits, motor, cognitive, at the electrophysiological level and in terms of certain cell biological variables (number of synaptic vesicles, gabaergic terminals).

We thank the referee for his/her comments.

1) Despite these interesting results, there are a number of questions that I would need the authors to answer before I can support the publication of this work in EMBO Mol. Med.

Does this treatment remain effective when applied for longer periods? This is of relevance as a) patients may need chronic treatment, and b) chronic exposure to cholesterol may be toxic to cells, especially if it accumulates in lysosomes.

We agree with the reviewer. In this work we used a transgenic mouse model of HD (i.g. R6/2 mice) with a strong and fast phenotype and an average life expectancy of around 13 weeks. In spite of the aggressivity of the model we show that the cholesterol treatment is efficacious. However, because of its rapid decline and death this mouse model does not allow to test chronic treatments. Furthermore, mini-pumps are not designed for very long treatments as the normal duration of infusion for ALZET pumps ranges from one day to six weeks, depending on the pump model, and replacement of the pump through a new surgical operation would be inappropriate in an animal that will not survive longer than 13 weeks.

Despite these limitations, we fully agree on the importance of this issue. For this reason, with the aim of translating cholesterol delivery to the clinic, we have been working for years on nanoparticles-mediated cholesterol delivery to the brain. Compared to our previous strategies (Valenza et al. 2015, EMBO Mol Med) we are now testing new nanoparticles (Belletti et al. 2018, Int J Pharm) with a higher cholesterol carrying capacity and capable of delivering a quantity of cholesterol to the brain similar to the most effective dose

identified with the mini-pump in this work. Hence, this presented work is a step forward in the direction of the long-term treatment requested by the reviewer. Furthermore, our ongoing trials with the new nanoparticles are performed in a knock-in animal model with a longer lifespan (12 months) and slower disease progression. We expect that having defined, in this current study, the dose of cholesterol needed to rescue HD phenotypes in R6/2 mice, the new study with nanoparticles in knock-in mice will fully achieve the target set by the reviewer and our studies.

2) This last point brings me to the next question: Where does the exogenous cholesterol partition? Authors should attempt to determine the membrane/intracellular localisation of the exogenous cholesterol (Bodipy-labelled cholesterol followed by LM?, biotinylated, non-cytolytic Perfringolysin O followed by EM?). It would be quite important to have more information on the localisation, especially to demonstrate that the exogenous cholesterol is present in the cellular domains where the endogenous cholesterol localises (plasma membrane, trans Golgi, late endosomes) and absent in ER and lysosomes, organelles where cholesterol accumulation is known to impair function.

This point is also important. In the original manuscript we have shown that exogenous bodipy-cholesterol infused with mini-pumps co-localizes with LAMP1, a marker for lysosomes, suggesting that exogenous cholesterol reaches these organelles. To gain more information about cholesterol partition, as suggested by the referee, we performed immunofluorescence analyses on coronal brain slices from R6/2 mice infused with bodipy-cholesterol to study the membrane/intracellular localization of the exogenous cholesterol. We used antibodies against TGN46, calnexin, Rab9A and PMCA-ATPase as markers for Golgi Apparatus, Endoplasmic Reticulum (ER), late endosomes and plasma membrane respectively. In the revised Figure EVII-O, we now show that bodipy-cholesterol does not localize with ER and Golgi but it partially co-localizes with late endosomes and plasma membrane.

We also evaluated the possibility to perform biotinylated, non-cytolytic perfringolysin O followed by EM, as suggested by the referee. However, to our understanding, this technique which is described in Mobius *et al.* 2002, doesn't guarantee the preservation of the ultrastructure of the tissue, while it works well in cells. Moreover, setting up the conditions for this experiment would require a very long time and a very precise expertise that unfortunately we do not have available.

3) Would the doses utilised in 7 week-old mice be equally effective in older mice? I am aware this question cannot be addressed experimentally with this strain of mice so I just ask authors to comment on this. This question is in my opinion pertinent for the simple reason that the average onset of this disease in humans is 40 years of age, which in the strain of mice utilised here would imply 10-14 month-old mice (not the 2 month-old mice used here).

As the referee pointed, with this HD mouse model we cannot test the efficacy of our strategy in older mice (please see our response in point 1). However, there are evidences in the literature indicating that aging is accompanied by the loss of cholesterol in the hippocampus leading to cognitive decline and altered short-term memory and learning (Martin *et al.* 2014, EMBO Mol Med; Martin *et al.* 2014, EMBO Rep). In the same studies, it was shown that the delivery of a very low dose of cholesterol in old mice through osmotic mini-pumps rescues the reduced hippocampal LTD associated with aging and improves hippocampal-dependent learning and memory in the water maze test (Martin *et al.* 2014, EMBO Mol Med). Finally, preventing cholesterol loss in old mice rescues BDNF transcription and improves cognition (Palomer *et al.* 2016, Cell Rep). We are therefore very hopeful that the dose used in this study will be equally effective in older mice and we are working exactly in this direction with the cholesterol-loaded nanoparticles (see point 1).

4) How long do the beneficial effects remain after infusion is terminated? This is an important aspect to be considered for future plans of applicability of this approach: continuous or discontinuous infusion? Did authors not allow some mice to live longer after the end of the 4 weeks of treatment?

As known to the reviewer and mentioned in point 1, R6/2 mice have a very severe phenotype and therefore we decided to avoid survival experiments. Moreover, leaving expired pumps *in situ* is not recommended as these pumps will swell and can leak concentrated salt solutions. We were also discouraged by our animal welfare body to remove the pumps at some point during the treatment as removal includes a surgical procedure in anesthetized animals and these animals are suffering. However, we agree with the referee that the duration of the beneficial effects after cholesterol supplementation is a crucial point. We believe that we will be able to address this point with the current study in knock-in mice described in point 1.

5) I also have some questions concerning the second objective of this work, on the mechanism involved in the improved phenotype. What is the contribution of increased endogenous cholesterol synthesis to the improved phenotype, compared to the effect of exogenous cholesterol? Co-infusion with an inhibitor of cholesterol synthesis (or a viral particle-mediated localised knockdown of SREBP2) could help to address this question. Although the (improved) phenotype due to cholesterol infusion is clear I consider important to address the contribution of the exogenous and endogenous cholesterol.

We thank the reviewer for the advice. The suggested experiment is very nice since it would allow to discriminate the effect of cholesterol infusion arising from the increased synthesis of the endogenous cholesterol from the contribution of the administered exogenous one. However, we have to consider that as synthesis of cholesterol in the brain of R6/2 mice is already strongly decreased, a further reduction of glial cholesterol production to levels below what is normally found in R6/2, may be too detrimental for HD neurons.

As a further comment, as shown in Figure 4B-E and Figure EV5, only the high dose of cholesterol led to an increase of 24S-OHC level (indirect measure of cholesterol catabolism) and of cholesterol precursors (indirect measure of cholesterol synthesis). Although all the tested doses of cholesterol equally prevented cognitive decline of R6/2 mice, suggesting a direct role of exogenous cholesterol on this parameter, motor defects were rescued only by the high dose, suggesting that a further stimulation of the endogenous synthesis of cholesterol and/or an increase of 24S-OHC level may be needed to have a beneficial effect on motor-related features.

6) The observation that exogenous cholesterol (Bodipy signal) colocalises with LAMP1 does not prove that exogenous cholesterol is responsible for the improved phenotype (see below).

We agree with the reviewer and we have now modified the text accordingly (see Discussion, pag. 22). We have now clarified that co-localization of LAMP1 and bodipy-cholesterol in the striatum of R6/2 mice (Figure EV10) suggests that exogenous cholesterol reaches lysosomes and may act on these organelles to enhance muHTT clearance.

7) How does increased biosynthesis occur in a high cholesterol content milieu? Authors assume that increased synthesis is due to high 24S hydroxycholesterol (from the exogenous cholesterol), in turn leading to SREPB2 activation in glial cells. An alternative view, even a simpler view, is that increased synthesis is the consequence of a further reduction of endogenous cholesterol due to cyclodextrin that becomes "free" in the injected area (once in the membrane cholesterol moves out of cyclodextrin ring), or direct activation of (endogenous) cholesterol catabolic activity (endogenous 24S OH Cholesterol) due to the presence of an ectopic (external) pool. Both conditions can result in reduced endogenous cholesterol (the increase 24 S hydroxyclosterol is from the endogenous pool) and consequently activation of SREBP2 pathway, which would be consistent with the canonical pathway of SREPB2 activation.

We thank the referee for these suggestions. The observation of an increased biosynthesis of cholesterol following the delivery of a high dose of cholesterol itself seems in fact counterintuitive. To test the hypothesis represented by the reviewer, we used isotopic dilution mass spectrometry (ID-MS) to measure the amount of cholesterol, lanosterol, lathosterol, desmosterol and 24S-OHC in the striatum of R6/2 mice treated with ACSF or methyl- β -cyclodextrin (M β CD) and previously tested for motor and cognitive tasks.

As shown in Figure EV2G-M, the level of cholesterol, cholesterol precursors and 24S-OHC, are not significantly changed in R6/2 mice following M β CD administration, indicating that M β CD, in this experimental paradigm, has no local effect on the endogenous synthesis/catabolism of cholesterol.

In our view, the most likely explanation is that an excess of exogenous cholesterol is metabolized to 24S-OHC which in turn leads to SREPB2 activation in glial cells via LXR stimulation and increased cholesterol biosynthesis. Cholesterol catabolism and synthesis are in fact closely related not only in healthy but also in diseased state: that is, more catabolism is accompanied by a greater synthesis (Shankaran *et al.*, 2017, Neurobiol Dis; Bossicault *et al.*, 2016, Brain). However, we can't exclude that other mechanisms may be at place. We modified the text in order to represent this level of uncertainty (see Discussion, pag 21).

8) It would be quite unique if cholesterol accumulation in lysosomes improves its degradative function: the Niemann Pick type C is a clear example of neurological (in addition to systemic) disorder due to accumulation of cholesterol in lysosomes. Therefore, the accumulation of cholesterol in LAMP1 organelles reported here is more likely synonym of lysosomal dysfunction. On the other hand, the result of reduced aggregated Huntingtin is clear, implying that improved degradation could be due to better degradative activity by the proteasome and/or phagosome.

It was not our intention to claim that cholesterol accumulation in lysosomes influences its function but only to report that (i) we do see cholesterol accumulation in lysosomes and that (ii) we do see reduced aggregated HTT as also highlighted by the reviewer. On this basis we can only hypothesize that the delivery of exogenous cholesterol, by reaching also the lysosomes, may contribute to stimulate/renormalize the autophagic flux likely by unblocking lysosomal accumulation.

There is extensive literature supporting this view and here we would like to mention only few of these reports. *First*, there is increased perinuclear accumulation of lysosomes in HD, which then leads to altered lysosomal-dependent functions and impaired autophagy (Jeong *et al.* 2010, Cell; Erie *et al.* 2015, Eur J Neurosci). *Second*, therapeutic strategies aimed at reducing lysosomal accumulation are able to reverse mutant HTT aggregation (Liang *et al.* 2011, Mol Neurodegeneration). *Third*, the chaperone-mediated autophagy (CMA), the mechanism responsible for protein degradation in lysosomes, resides in microdomains of the lysosomal membrane enriched in cholesterol and glycosphingolipids (Kaushik *et al.* 2006, The EMBO Journal). *Fourth*, lysosomal membranes from 18QHtt and 111QHtt knock-in HD mice show a significant reduction in cholesterol in the HD genotype (Koga *et al.* 2011, J. Neurosci). Thus, strategies aimed at restoring concentration/distribution of cholesterol in lysosomes may have a positive effect on the activity of key lysosomal proteins and on the clearance of aggregated proteins.

We also agree with the reviewer that activation of proteasome and/or phagosome may contribute to increase mutant Huntingtin clearance.

9) Authors should do more to determine how exogenous cholesterol increases degradation (does the increase in lanosterol enhance mTORC1 activity?).

We understand well the point raised by the reviewer. However, we believe that exploring the link between mTORC1 and cholesterol will require several months of new experiments that would imply a totally new project. We are now testing this link in neurons differentiated from human induced pluripotent stem cells carrying HTT with 21Q or 109Q. We treated wt and HD neurons for 5 days (from day25 to day30) with 10 μ g/mL of cholesterol followed by western blot analysis to evaluate the activity of mTORC1. By analyzing the ratio between phosphorylated ribosomal protein (pS6) and S6, we see that mTORC1 activity is decreased in HD neurons and that this phenotype is reversed by cholesterol supplementation.

Although promising, we believe that further studies are needed to confirm this result and deeply explore the link between cholesterol and mTORC1 in an HD background. Of note, these findings are also in agreement with a previous paper demonstrating that mTORC1 activity is reduced in the striatum of HD mice and its activation enhances cholesterol biosynthesis genes and counteracts muHTT aggregation by stimulating clearance pathways (Lee *et al.* 2015, Neuron).

10) Authors should be very cautious associating improved lysosomal function with cholesterol accumulation (quoting authors: Colocalization of LAMP1 and bodipy in the striatum of R6/2 mice (Fig 6E) suggests that cholesterol may act directly on lysosomes to enhance muHTT clearance)

We take the point raised and have modified the text accordingly (see Discussion, pag 22). We also would like to stress that we did not demonstrate that there is accumulation of exogenous cholesterol in lysosomes, but that exogenous cholesterol localizes with LAMP1, a marker of these organelles (see Figure EV10), where it can have some function.

REFEREE #2 (REMARKS FOR AUTHOR):

This is an interesting and thorough manuscript done by an expert team. The experiments are well designed and the conclusions are meaningful. I have a number of concerns regarding interpretations and conclusions which should be addressed.

We thank the reviewer for his/her comments.

1) Figure 1D: I am concerned with the baseline Rotarod performance of the mice at 5 weeks (preimplantation). Although not statistically significantly different, the baseline performance of the mice destined to be in the high dose group is greater than the control R6/2 mice at a magnitude which approximately persisted throughout the trial. What were the starting weights of the R6/2 mice in the different groups at the outset and through the trial? Were they the same sex? Were there any differences in end-stage or mortality?

We thank the referee for this observation. We had noted that the baseline Rotarod performance of the mice destined to be in the high-chol group is greater than the control R6/2 mice. We think that these differences, that are not statistically significant, may be due to chance and to some variability of the animals at this early time point. Importantly, no differences have been measured in the rotarod performance at 5 weeks of age between wt and all the R6/2 mice groups, indicating that this is still a pre-symptomatic time point.

With respect to starting weights and sex of the mice in the different groups, in the revised version of the manuscript (Figure EV3) we have now included the body weight of all mice used in the behavioral trials. In all trials the animals were assigned randomly, and sex was balanced in the various experimental groups. The body weight was similar across groups at 7 weeks of age, before surgery (Figure EV3D-E). However, at 12 weeks of age, all R6/2 males showed a reduced body weight compared to wt, with the only exception of the R6/2 chol-high group whose body weight was similar to that of wt mice.

2) Figure 2A-B: From the methods, the cannula was inserted in the right hemisphere and the studies were done on the right side. Given the remarkable effects noted on figure 1, a question is why the magnitude of the behavioral data was as profound as it was while the infusion and biochemical effects were only unilateral? It appears from the data in supplementary figure 3 that there is no increase in

cholesterol levels in the contralateral hemisphere. How do the authors explain the remarkable impact given that HD is a bilateral disease and the therapy and the biochemical changes are unilateral?

We asked ourselves as well about this point since exogenous cholesterol clearly remains only in the infused hemisphere. We hypothesize that functional rescue of neural circuits in the unilaterally infused striatum of R6/2-chol mice is sufficient to perform some tasks correctly. This is evident in the NOR test, in which the ability to recognize a new object compared to a familiar one may under unilateral control and the restoration of circuits in one hemisphere may therefore compensate for global (bilateral) deficits. In other words, it is reasonable to expect that the correct information of memory recognition processed by the infused striatum of HD mice and the further elaboration of this information in the decisional areas of the ipsi-lateral prefrontal cortex may be sufficient to have a positive outcome of the NOR test.

There are examples of this unilateral recovery also for motor system. For example, in Parkinson Disease animal models unilateral manipulations of one nigrostriatal system was found to affect contralateral dopamine turnover, indicating a functional and compensatory inter-dependence of the two nigrostriatal systems (Blesa *et al.*, 2011 *Front. Syst. Neurosci.*). And it is on this basis that the success of unilateral injection of fetal ventral mesencephalon-based cell therapy in Parkinson's Disease patients can be explained (Li *et al.*, 2016 *PNAS*). The improvement we see in motor tasks with the higher cholesterol dose may fall into the condition in which a certain level of a unilateral therapy may induce global improvement. However, we acknowledge that more work is needed in order to fully address this point.

3) Figure 2C: Please clarify in the figure to which group is the middle bar different, WT and/or high Chol?

Apologies but we do not understand this comment. In Figure 2C there are no differences between the groups.

4) Figure 4F: The morphology of stained cells appear potentially as neuronal. Please perform stain with neuronal and astrocytic marker in these slices. The cells demonstrating nuclear translocation look more like the cells in 4H than the ones in 4I. Please show what a WT control slice looks like.

We thank the referee for this note. The localization of SREBP2 is peri-nuclear (as inactive form) and nuclear (as active form), thus it is very difficult to discriminate between neurons and astrocytes without a specific cell marker. As suggested by the reviewer, we performed staining for SREBP2 with neuronal and astrocytic markers (NeuN and GFAP respectively) in coronal brain slices from R6/2 mice treated with the high dose of cholesterol. In the original Figure 4F, we used an old aliquot of SREBP2 antibody obtained from Dr. T. Osborne (Johns Hopkins University, Baltimore, Maryland, USA; described in Seo, *Cell Metabolism* 2012). This antibody works well but it is not commercially available. In August 2019, we obtained from Dr. Osborne another aliquot of the same antibody and performed immunostaining for SREBP2 with NeuN and GFAP. We showed the different localization of SREBP2 in astrocytes (GFAP) and neurons (NeuN) in R6/2-chol mice. However, the staining was suboptimal.

To confirm further the perinuclear/nuclear staining of SREBP2, we decided to use another antibody. We first tested four commercial antibodies in brain slices of wt mice (R&D, AF7119; Abcam, ab30682; LsBio, rabbit, LS-S4695; LsBio, mouse, LS-C179708). The latter (LsBio, mouse, LS-C179708) was the most promising and, with this antibody, we performed immunofluorescence staining of SREBP2 in brain slices of wt, R6/2 and R6/2-chol mice. Accordingly, we modified Figure 4 by replacing the old images with those obtained with the new antibody from LsBio. As shown in the revised Figure 4, SREBP2 localization is nuclear and perinuclear in the striatum of wt mice (Figure 4F-G). On the contrary, SREBP2 nuclear localization is reduced in the striatum of untreated R6/2 mice (Figure 4F-G) compared to wt mice. Of note, nuclear translocation of SREBP2 is increased in the infused striatum (Figure 4H-I) compared to the contralateral striatum of R6/2-chol mice (Figure 4H-I), confirming the findings obtained with the previous antibody gifted by Dr. Osborne.

We also performed co-staining of SREBP2 with neuronal (NeuN) and astrocytic (GFAP) markers in the infused striatum of R6/2-chol mice. The morphology of cells stained with SREBP2 antibody is similar; however, the nuclear localization of SREBP2 is appreciable only in astrocytes (Figure 4L-M), in agreement

with the notion that in the adult brain cholesterol synthesis (and nuclear distribution of SREBP2) occurs mainly in this cell type (Mauch *et al.* 2001, Science; Camargo *et al.* 2012, Faseb; Ferris *et al.* 2017, PNAS).

5) What would a WT slice look with cholesterol infusion (i.e. translocation)?

Unfortunately, we do not have coronal brain slices from wt mice treated with cholesterol. As we did not see differences in terms of behavior, we collected tissues only to perform mass spectrometry analysis. However, as we found an increase of cholesterol precursors and 24S-OHC also in wt mice treated with the high dose of cholesterol (Figure EV5A, C, E, G), we can speculate that striatal infusion of cholesterol would lead to an increase of SREBP2 nuclear translocation also in the healthy brain. Of note, the behavior in wt mice is not influenced by cholesterol treatment (Figure EV1A-F) suggesting that compensatory mechanisms occur to not influence motor or cognitive performance. In HD mice, where cholesterol synthesis and catabolism are reduced, the striatal infusion of cholesterol likely stimulates the same compensatory mechanisms that lead, in this case, to a normalization of these pathways with a positive benefit on the behavior.

6) The data in figure 5 is confusing. EM48 staining is reduced in the infused region. It is my understanding that the pattern of EM48 staining represents the mHTT aggregates. The data in figure 5N and O points to the soluble as the mHTT species that is impacted. Please explain and clarify in the text.

Mutant Huntingtin (muHTT) aggregation occurs with a complex and multi-step mechanisms as the protein undergoes post-translational modifications, leading to abnormal conformations. Mutant Huntingtin forms oligomer intermediates that then give rise to globular intermediates from which protofibrils are generated. Protofibril intermediates associate to produce amyloid-like structures, resulting in macro-aggregates or inclusions. There are different assays to measure and analyze different stages of mutant Huntingtin aggregation and different muHTT species. Here are the ones we have used:

First, immunofluorescence staining on brain sections by using the EM48 antibody, which is specific for the expanded polyQ tract prone to aggregate, allow to measure macro-aggregates of around 2 μm . With this assay, we revealed that the number and the size of muHTT aggregates is reduced in the infused striatum of chol-high mice compared to control (Figure 5A).

Second, electron microscopy allows to obtain additional information about the morphology of muHTT aggregates and explore muHTT species that aggregate as protofibril-like structures of about 300 nm or dispersed in the cytoplasm and nucleus of striatal neurons from HD mice (as we also see in R6/2-ACSF mice). In contrast, muHTT was found dispersed and never composed in a fibril network in striatal neurons from R6/2 chol-high animals (Figure 5H).

Finally, time-resolved Förster resonance energy transfer (TR-FRET)-based immunoassay (Figure 5I) is helpful to explore the early phases of aggregation process as it allows quantification of muHTT oligomers that tend to aggregate at the beginning of the aggregation process. The use of 4C9-4C9 combination of antibodies against HTT allows to discriminate muHTT oligomers from muHTT that includes soluble muHTT and other muHTT species (detected with the 2B7-MW1 combination).

Our findings indicate that exogenous cholesterol (i) influences different “late” stages but not the “early” stages of aggregation process (as levels of muHTT oligomers are not changed) and (ii) reduces muHTT aggregates by increasing the clearance of the protein rather than by dissolving the various aggregated forms of the protein. We have modified the text to clarify this point (see Results, pag. 17).

7) As noted in the discussion, demonstrating a link between the therapy and mTORC1 would add mechanistic insight to the study.

Please see our response to referee #1 (point 9). We do believe this is an important issue. However, we think that several experiments are required to explore this link.

REFEREE #3 (REMARKS FOR AUTHOR):

This study examined the effects of infusion of cholesterol into the striatum of HD R6/2 mice to determine if cholesterol deficits and phenotypes could be altered. The authors found increase in cholesterol synthesis, improvement in cognitive and motor behaviors and synaptic activity, and reduction in aggregates and mutant huntingtin levels. The work builds on earlier findings by some of the authors, is very carefully executed, and very nicely presented. These data in mice have considerable significance for considering cholesterol supplementation as a potential therapy for HD patients and will be of a great interest to a wide scientific community.

We thank the referee and we appreciate these comments.

1) The interpretation of some results is weak as pointed out below. The underlying mechanisms for how elevated cholesterol in astrocytes improves neuronal function in different subcellular compartments is loosely connected.

We apologize for not being clear. We added more information in the text to clarify this point (see Introduction, pag. 3 and Discussion, pag. 21).

2) Fig 1. The behavioral benefits of cholesterol infusion were achieved using ipsilateral delivery to the striatum. This is somewhat surprising. The authors should comment on why unilateral infusion was sufficient and if bilateral infusion would have allowed for a lower dose to be used for many of the studies.

Please, see our response to referee #2 (point 2).

It is definitely possible that the use of bilateral infusion of cholesterol would have allowed to reach the same benefit with lower doses of cholesterol. We avoided bilateral infusion, which can be achieved by connecting a single ALZET pump to a Y connector, because the caveat of using the such connector is that even distribution of the solution between the two outputs is not guaranteed. To ensure accurate and even bilateral distribution, the guidelines of ALZET recommend implanting two pumps simultaneously, with each pump connected to a catheter leading to each target site at the corresponding brain hemisphere. However, implanting more than one ALZET pump is only feasible in animals large enough to accommodate the additional size and weight of multiple pumps and this approach is not recommended in R6/2 mice.

3) Fig. 1 Corpus callosum contains some fluorescent cholesterol and cortex was also infused. How do the benefits attributed to striatum relate to improved corticostriatal connectivity particularly excitatory activity? Were other brain areas also fluorescent?

As visualized by the bodipy-cholesterol spread, exogenous cholesterol remains only in the infused hemisphere. Only few cells of the cortex in the ipsi-lateral hemisphere were found fluorescent (probably because of cholesterol spread out the cannula during the insertion in the striatum) and no other brain areas were fluorescent. However, by quantifying cholesterol content by mass spectrometry, we found a significant increase of cholesterol also in the ipsi-lateral cortex of HD mice (Figure EV3A), suggesting that cortical neurons containing cholesterol may also contribute to the functional recovery of striatal target cells. Of note, the electrophysiological analysis was performed throughout the entire right striatum, so the probability to record a specific striatal cell that is connected to a specific positive cortical neuron is very low. In light of this consideration, it is more likely to assume that cholesterol exerts most of its benefits directly in the striatum, and in particular at the level of the synaptic contacts between the axon of a cortical neuron and a striatal MSN - even if we cannot exclude also a direct effect of cholesterol in a small subpopulation of cortical neurons.

4) Fig 2G. The total number of excitatory synapses based on FIB-SEM was not changed by cholesterol infusion.

Correct

5) Fig 2H-I. The analysis of synaptic vesicle depletion in HD and increase of synaptic vesicles after cholesterol infusion should consider other variables. Did the authors take into account the total number of synaptic vesicles, length of the synaptic contact and the number of vesicles at non synaptic membranes in the terminal as well?

We apologize for not being clear in our original manuscript and for the missing information. We have now improved this part of the study by performing new analysis. We quantified the pre-synaptic area (μm^2) and the active zone length (μm). Analysis of presynaptic structures did not reveal differences in terms of bouton morphology between wild-type and R6/2 mice with or without cholesterol treatment. We have now included this analysis in Figure EV4C, D and we modified the text accordingly (see Results, pag 11).

Regarding the analysis of the total number of synaptic vesicles we thank the reviewer for his/her observation. We reported the analysis of the density of the synaptic vesicles (SVs) in figure 2L in the original manuscript. We quantified the SVs density as the total number of SVs divided by the pre-synapse area (μm^2). We apologize if this was not clear; we have added now a sentence in the Results (pag. 11) and in the Material and Methods (pag. 33) to stress this point.

6) If these are excitatory synapses, the synaptic contact should look asymmetric. The R6/2 example (upper right) does not look asymmetric in the EM image as it is depicted in the adjacent drawing.

We thank the reviewer for his/her suggestion. Accordingly, the R6/2 images in Figure 2H-I have been changed in the revised version of the manuscript.

7) The scale bar for H and I seems off and should be checked. Synaptic vesicles are typically about 30-50 nm.

We thank the reviewer for his/her observation. We corrected the scale bar in Figure 2H-I.

8) Results for Fig 2 G, Fig 3, Fig 4, and Fig 5 are not dose dependent and therefore somewhat misleading given the title of the paper.

We do agree with the reviewer. Accordingly, we changed the title to “Striatal infusion of cholesterol promotes dose-dependent behavioral benefits and exerts disease-modifying effects in Huntington’s disease mice”

9) Fig 4. It is not clear how increase in cholesterol in astrocytes affects neurons.

We apologize for not being clear. We added more information in the Introduction to clarify this point (pag. 3).

In the adult brain, cholesterol is produced by astrocytes and supplied to neurons for their activities (Pfrieger & Ungerer 2011, Prog. Lipid. Res). This mechanism is impaired *in vitro* in the presence of the mutation. Accordingly, in CSF from HD mice ApoE was associated with smaller lipoprotein particles (Valenza *et al.* 2010, J. Neurosci), suggesting that HD astrocytes display reduced cholesterol biosynthesis and efflux (Valenza *et al.* 2010). Furthermore, it has been shown that glia-conditioned medium (GCM) from wt astrocytes rescued neurite outgrowth defect in HD neurons. This beneficial effect of wt GCM was abolished by disrupting cholesterol production in wt astrocytes (Valenza *et al.* 2015, Cell & Death Differentiation). On the contrary, GCM derived from HD astrocytes failed to support neurite outgrowth in HD neurons. However, the stimulation of cholesterol synthesis in HD astrocytes led to increased ApoE/cholesterol release in the medium reversing neurite outgrowth and synaptic defects in HD neurons (Valenza *et al.* 2015, Cell & Death Differentiation).

10) The legend and labeling do not agree. DARPP32 is not labeled in the figure.

Corrected. Thanks.

11) Fig 5 F Astrocyte size and branching looks increased in the image of the high cholesterol infused side. Are the cells reactive? Has cholesterol increased GFAP levels? Are microglia reactive?

As the reviewer highlighted, astrocytes in the infused striatum seems to be active. Probably, the brain infusion kit connected to the mini-pumps created a small damage that led to astrogliosis. Since this phenomenon does not affect the behavior and the health status of R6/2 mice, we did not check GFAP levels and the microglia reactivity.

12) Fig 5H legend describes EM 48 in the striatal neuropil but only cell bodies are shown.

We agree. We have now clarified this point in the legend of Figure 5H as suggested.

13) In the R6/2 ACSF treatment in left column images -the fibrous structure described near the nuclear membrane is not clear but underneath that labeling fibrillar structure is more apparent.

We apologize for not being clear in our original manuscript. We added a sentence in the legend of Figure 5H.

14) Fig 5N. There is no change in aggregate load with TR FRET 4C9-4C9 aggregate assay with high dose cholesterol infusions compared to marked reduction in aggregates seen by immunostaining. What accounts for this apparent discrepancy? Small sample size or the assay itself?

As described above (see point 6 to referee #2), muHTT aggregation occurs via a multi-step mechanism and different assays are available to measure and analyze different muHTT species at different stages of aggregation. Accordingly, TR-FRET (4C9-4C9) assay detects and quantifies muHTT oligomers prone to aggregate at the beginning of the aggregation process, which are not detectable with EM48 immunostaining. As the referee highlighted, with this assay we did not see changes in aggregated muHTT suggesting that exogenous cholesterol does not influence the early stages of aggregation process (Figure 5N) while it is able to influence the other aggregated forms of the mutant protein (Figure 5A-H).

15) Fig 6. The presence of perinuclear Lamp1 lysosomes is interpreted as clearance by lysosomes. It is not clear what they are clearing. The authors should consider the other possibility that the HD mutation impairs anterograde transport that results in retrograde accumulation of lysosomes. Depletion of synaptic vesicles may also be due to impaired anterograde transport.

We thank the referee for her/his comment. We modified the text to consider also this possibility (see Discussion, pag. 22).

16) High dose cholesterol did not rescue myelin depletion in the R62 mice. Were oligodendroglia markers checked?

We did not check oligodendroglia markers. Myelination occurs in a very early stage of the life. Furthermore, in the brain of P14 R6/2 mouse model there is a significant decrease in MBP staining intensity in the corpus callosum and also in the striatum (Xiang *et al.* 2011, J. Neurosci). In mice, the highest cholesterol synthesis rate occurs during the first postnatal weeks, concomitantly with the peak of the myelination process. After myelination, the metabolism of cholesterol in the adult brain is characterized by a very low turnover, with estimated half-life of 2–6 months in rodents and 5 years in humans (Jurevics & Morell 1995, J. Neurochem; Bjorkhem *et al.* 1998, J. Lipid Res). Probably, in our experiments the delivery of cholesterol in adult mice (from 7 weeks of age) occurs too late to counteract this defect.

17) Authors suggest from their data that there is increased availability of cholesterol in glial cells. How does this in turn enhance neuronal function? Could the cholesterol upload by glia be a reactive/phagocytic response?

In this work, we demonstrated that the delivery of a high dose of cholesterol leads to increased levels of 24S-OHC, the neuronal-specific metabolite of cholesterol. It is known from the literature that neuronal 24S-OHC induces apoE transcription, protein synthesis, and secretion in a dose- and time-dependent manner to supply cholesterol to neurons (Abildayeva *et al.* JBC 2006). We hypothesize that in R6/2-chol mice, the increased level of 24S-OHC stimulates cholesterol efflux from HD astrocytes, that, in turn, increases endogenous cholesterol synthesis. However, we can't exclude that other mechanisms are involved.

Of note, our previous findings showed that even if cholesterol synthesis is reduced in HD astrocytes, this cell type is still able to maintain similar level of intracellular cholesterol compared to wt astrocytes by diminishing cholesterol efflux (Valenza *et al.* 2010, J. Neurosci; Valenza *et al.* 2015, Cell & Death Differentiation). This suggests that HD astrocytes may be able to manage cholesterol upload in the absence of reactive/phagocytic response.

7th Aug 2020

Dear Dr. Valenza,

Thank you for the submission of your revised manuscript to EMBO Molecular Medicine. We have now received the enclosed reports from the referees that were asked to re-assess it. As you will see the reviewers are now globally supportive and I am pleased to inform you that we will be able to accept your manuscript pending the following final amendments:

***** Reviewer's comments *****

Referee #1 (Remarks for Author):

For the majority, authors have responded satisfactorily to my comments and suggestions. My recommendation is that some aspects of the answers to comments 1 and 3 are included in the discussion of this work. The very short reference to the second generation of nanoparticles at the end of discussion should be used to reinforce the idea that this might be a strategy to use in the future, in long-term therapeutic approximations.

On the other hand, the authors should re-consider their idea that phenotype improvement is due to higher cholesterol content in lysosomes. The results of these authors and many others leave little doubt that HD phenotype improves by improving lysosomal function. However, this does not imply that the improvement in phenotype shown in this work is due to higher levels of cholesterol in lysosomes. I am sure the authors know very well that the presence of lipids such as SM, cholesterol or ceramide in the lysosomal membrane interferes with the activity of saponising enzymes and therefore the degradation/uncoupling of cargo/es. And that in order to guarantee that endocytic material reaches the lysosomes with a minimum of cholesterol endocytic vesicles / organelles carry cholesterol-removing enzymes: NPC1, NPC2, ABCA1. I repeat: the authors should be more open to the possibility that the beneficial effect of increasing cholesterol is through a mechanism other than the accumulation of cholesterol in lysosomes.... (on this line, I recommend to consider removing/replacing the sentence " Of note, we showed here that exogenous cholesterol localizes in lysosomes and may reduce lysosome accumulation, leading to muHTT clearance". I find this conclusion too far fetched, not supported by the data.

Referee #2 (Remarks for Author):

I appreciate the thorough revision, clarifications and new information. Exciting manuscript.

Referee #3 (Remarks for Author):

This reviewers concerns have been satisfactorily addressed.

Referee #1 (Remarks for Author):

For the majority, authors have responded satisfactorily to my comments and suggestions. My recommendation is that some aspects of the answers to comments 1 and 3 are included in the discussion of this work. The very short reference to the second generation of nanoparticles at the end of discussion should be used to reinforce the idea that this might be a strategy to use in the future, in long-term therapeutic approximations.

On the other hand, the authors should re-consider their idea that phenotype improvement is due to higher cholesterol content in lysosomes. The results of these authors and many others leave little doubt that HD phenotype improves by improving lysosomal function. However, this does not imply that the improvement in phenotype shown in this work is due to higher levels of cholesterol in lysosomes. I am sure the authors know very well that the presence of lipids such as SM, cholesterol or ceramide in the lysosomal membrane interferes with the activity of saponising enzymes and therefore the degradation/uncoupling of cargo/es. And that in order to guarantee that endocytic material reaches the lysosomes with a minimum of cholesterol endocytic vesicles / organelles carry cholesterol-removing enzymes: NPC1, NPC2, ABCA1. I repeat: the authors should be more open to the possibility that the beneficial effect of increasing cholesterol is through a mechanism other than the accumulation of cholesterol in lysosomes.... (on this line, I recommend to consider removing/replacing the sentence " Of note, we showed here that exogenous cholesterol localizes in lysosomes and may reduce lysosome accumulation, leading to muHTT clearance". I find this conclusion too far fetched, not supported by the data.

We thank the referee for the suggestions. We modified the last paragraph of the discussion in order to include briefly answers to comments 1 and 3: "Further studies will explore the potential for long-term cholesterol release in HD animal models with a longer lifespan and slower disease progression, enabling chronic treatment in older, symptomatic mice. In addition, with the aim of translating the delivery of cholesterol to the clinic, new brain-permeable nanoparticles have been developed (Belletti et al, 2018) that enable the controlled release of a higher cholesterol content to the brain. This advance may facilitate progress toward the goal of achieving the therapeutic dose identified here by systemic injection". As suggested, we also modified the discussion paragraph by removing the sentence "Of note, we showed here that exogenous cholesterol localizes in lysosomes and may reduce lysosome accumulation, leading to muHTT clearance".

Referee #2 (Remarks for Author):

I appreciate the thorough revision, clarifications and new information. Exciting manuscript.

We thank the referee #2.

Referee #3 (Remarks for Author):

This reviewer concerns have been satisfactorily addressed.

We thank the referee #3.

The authors performed the requested changes.

Corresponding Author Name: Elena Cattaneo, Marta Valenza

Manuscript Number: EMM-2020-12519